# Tumour-associated missense mutations in the dMi-2 ATPase alters nucleosome remodelling properties in a mutation-specific manner

Kristina Kovač [1], Anja Sauer[1], Igor Mačinković[1], Stephan Awe[1], Florian Finkernagel [1,2], Helen Hoffmeister[3], Andreas Fuchs [3], Rolf Müller[1,2], Christina Rathke[4], Gernot Längst[3] & Alexander Brehm[1]

ATP-dependent chromatin remodellers are mutated in more than 20% of human cancers. The consequences of these mutations on enzyme function are poorly understood. Here, we characterise the effects of CHD4 mutations identified in endometrial carcinoma on the remodelling properties of dMi-2, the highly conserved *Drosophila* homologue of CHD4. Mutations from different patients have surprisingly diverse defects on nucleosome binding, ATPase activity and nucleosome remodelling. Unexpectedly, we identify both mutations that decrease and increase the enzyme activity. Our results define the chromodomains and a novel regulatory region as essential for nucleosome remodelling. Genetic experiments in *Drosophila* demonstrate that expression of cancer-derived dMi-2 mutants misregulates differentiation of epithelial wing structures and produces phenotypes that correlate with their nucleosome remodelling properties. Our results help to define the defects of CHD4 in cancer at the mechanistic level and provide the basis for the development of molecular approaches aimed at restoring their activity.

[1] Institute for Molecular Biology and Tumour Research, University of Marburg, 35043 Marburg, Germany. [2] Center for Tumour Biology and Immunology, University of Marburg, 35043 Marburg, Germany. [3] Institute of Biochemistry, Genetics and Microbiology, University of Regensburg, 93053 Regensburg, Germany. [4] Department of Biology, Philipps-University Marburg, Karl-von-Frisch-Straße 8, 35043 Marburg, Germany. Correspondence and requests for materials should be addressed to A.B. (email: brehm@imt.uni-marburg.de)

Eukaryotic cells use three enzyme classes to generate an epigenome appropriate for their cell type and state: DNA methyltransferases and demethylases that regulate CpG methylation, histone modifying enzymes that add or remove chemical groups from histones and nucleosome remodellers that couple ATP hydrolysis to alterations in nucleosome structure and position. Aberrant DNA methylation and histone modification states are a hallmark of many cancers, and inhibitors of DNA methyltransferases, histone deacetylases and histone methyl-transferases are used in cancer therapy[1]. Much less is known about the molecular basis underlying defects in nucleosome remodellers and their contribution to a diseased state.

Mutations in subunits of nucleosome remodelling complexes are associated with over 20% of human cancers[2,3]. CHD4, the ATPase subunit of the Nucleosome Remodelling and Deacetylase (NuRD) complex, is frequently mutated in endometrial carcinoma[4-7]. Endometrial carcinoma of the serous type has the highest incidence of CHD4 mutations (17%) but CHD4 mutations are also present in mixed histology, endometrioid and clear cell carcinoma. In addition, CHD4 mutations are associated with several other cancers, with thyroid cancer, ovarian cancer and lymphoma most frequently affected[2]. CHD4 mutations are not restricted to cancer and have also been identified in patients with intellectual disability syndromes[8].

CHD3 and CHD4 are subunits of alternative NuRD complexes[9]. NuRD complexes regulate genes by modulating their expression levels positively or negatively[10]. Recent work has highlighted the importance of the CHD4 remodeller for NuRD-mediated repression of lineage-specific genes during differentiation in both ES cells and B-cells[10-12]. CHD4-mediated nucleosome remodelling is required for the early steps of establishing a repressive chromatin structure. These early steps are characterised by a rapid increase in nucleosome density and the loss of RNAP2 and activating transcription factors. Changes in histone modifications, such as deacetylation, and intranuclear relocalisation to heterochromatic regions occur later and stabilise the long-term repression of target genes. Importantly, these later steps depend on the preceding nucleosome repositioning catalysed by CHD4. These results imply that mutations in CHD4 that affect its nucleosome remodelling function may have profound effects on gene expression.

However, it is not known whether and how disease-associated CHD4 mutations affect its enzymatic activities at the molecular level and how this impacts the epigenetic landscape, gene expression and genome stability. Analysis of the CHD4 mutation spectrum reveals two remarkable features: First, the majority of CHD4 alterations are missense mutations (89% in endometrial carcinoma)[4,6,7]. Deletions, frameshift and nonsense mutations that would result in a complete loss or a truncated CHD4 protein are rare. Second, patients are heterozygous for CHD4 missense mutations and retain one wild-type (WT) copy of CHD4[4,6,7]. It is possible that mutations in one of the two dMi-2 alleles lowers the total CHD4 activity in affected cells sufficiently to result in the misregulation of genes. Alternatively, CHD4 point mutants might exert a dominant negative effect or result in a gain of function.

CHD4 missense mutations identified in endometrial cancer are enriched in the ATPase domain (24 missense mutations affecting 19 residues, Fig. 1). Four mutated residues map to the PHD finger and double chromodomain regions. Eleven missense mutations map to the region C-terminal to the ATPase domain, three of which are within 100 amino acids of the catalytic domain. By contrast, the N-terminal region preceding the PHD finger/chromodomain region is largely devoid of missense mutations.

Recent elegant functional and structural analyses of nucleosome remodellers from different model organisms have dramatically improved our understanding of how these machines work at the molecular level[13-16]. Together these studies have provided two key insights. First, the regions outside the catalytic domain regulate the ATPase motor by directly contacting its catalytic core and/or the nucleosome substrate[17]. Such regulatory regions have been identified in yeast Chd1 (chromo wedge), yeast and Drosophila ISWI (AutoN and NegC domains) and yeast Snf2 (brace-I and brace-II helices)[13,18-21]. Second, the single point mutations not only result in a dramatic loss of enzymatic activity but can also lead to increased remodelling activity or produce complex outcomes affecting the coupling of ATP hydrolysis and nucleosome remodelling[19,22]. These insights make it difficult to predict the consequences of a missense mutation based on where it is located in the remodeller (within or outside of the ATPase domain). Moreover, even for mutations within the catalytic core, it is difficult to predict if they have a negative impact, a positive impact or no impact on remodelling activity. It is, therefore, essential to determine experimentally the molecular consequences of disease-derived missense mutations in nucleosome remodellers to gain a better understanding of how they contribute to disease.

Here, we report a systematic analysis of cancer-derived CHD4 missense mutations on nucleosome remodelling. We introduced selected point mutations into the PHD fingers, chromodomains, ATPase domain and the adjacent C-terminal region of the highly conserved Drosophila CHD4 homologue dMi-2 and tested the capacity of mutants for ATP, DNA and nucleosome binding. We also determined their nucleosome-stimulated ATPase activity, their ability to make nucleosomal DNA accessible to restriction enzymes and to slide nucleosomes along linear DNA fragments. Our results uncover a remarkable breadth and variety of effects. We identify mutations that decrease or even increase nucleosome remodelling activity to varying degrees. Mechanistically, we characterise mutations that affect nucleosome binding, ATPase activity and/or the coupling of ATP hydrolysis and nucleosome positioning. We reveal a novel regulatory region outside of the ATPase domain that is essential for nucleosome remodelling. Furthermore, our results provide support for a recently proposed model suggesting that the chromodomains of CHD enzymes directly contact nucleosomal DNA during the remodelling

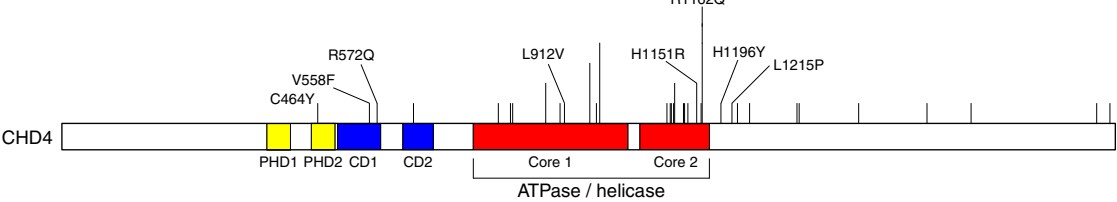

**Fig. 1** CHD4 missense mutations identified in endometrial cancer. Fifty-two reported CHD4 mutations were taken into account[4,6,7]. Mutations identified in samples with a hypermutation phenotype (multiple mutations in CHD4 and/or mutations in DNA polymerase epsilon) were disregarded. Locations of individual missense mutations are shown by a vertical line. The length of this line represents the frequency with which the affected residue was found mutated. Missense mutations analysed in this study are depicted on top. PHD PHD finger, CD chromodomain

reaction. Endometrial cancer is caused by transformation of epithelial cells. Therefore, we used a *Drosophila* model for epithelial differentiation to assess the impact of cancer-derived dMi-2/CHD4 mutants in vivo. We demonstrate that expression of dMi-2 mutants is sufficient to change cell fate in the developing wing. Taken together, our results provide first mechanistic insights into the defects of dMi-2/CHD4 nucleosome remodellers in cancer.

## Results

**Cancer-derived mutations in CHD4.** The majority of missense mutations identified in endometrial cancer map to the three domains of CHD4: the PHD fingers, the double chromodomains and the ATPase domain (Fig. 1). The mutated residues are mostly conserved between human CHD4 and related enzymes of other species. Indeed, of 37 mutated residues 34 are identical in the *Drosophila* CHD4 homologue dMi-2.

In the past, we have used dMi-2 as a paradigm to characterise the nucleosome remodelling properties of CHD enzymes in vitro[23–27] and to analyse the role this remodeller in development and differentiation[28–31]. Here, we introduced seven endometrial cancer-derived missense mutations into dMi-2 by site-directed mutagenesis (Fig. 1). In addition, we have identified a mutation in CHD4 using previously published RNAseq data from the ovarian carcinoma cell line SKOV-3 (H1196Y) which we have included in the analysis[32]. All eight residues are identical between human CHD4 and dMi-2 and reside in regions with a high degree of sequence similarity (Supplementary Fig. 1A). To avoid confusion, we will use the amino acid numbering of human CHD4 also when referring to mutations introduced into dMi-2. The corresponding amino acid positions of CHD4 and dMi-2 are given in Supplementary Fig. 1B. For each mutant, we generated recombinant baculoviruses and purified proteins from extracts of infected Sf9 cells (Supplementary Fig. 1C).

**dMi-2 mutants bind ATP.** We first tested the ability of dMi-2 mutants to bind to ATP using a filter binding assay. Binding of WT dMi-2 to radioactively labelled ATP was efficiently competed by an excess of cold ATP but not by an excess of cold GTP, verifying specificity of nucleotide binding in this assay (Supplementary Fig. 1D). Most dMi-2 mutants displayed robust ATP binding activity (60–150% of WT ATP binding activity, Supplementary Fig. 1E). ATP binding of dMi-2 R572Q was more strongly reduced but even this mutant retained 32% of WT binding activity. In agreement with these findings, superimposition of the dMi-2 sequence onto the yeast Chd1 structure revealed that none of the mutated residues is in close proximity to the ATP binding pocket[13].

**Mutations in the ATPase domain decrease remodelling activity.** We next assessed nucleosome binding, ATPase and nucleosome remodelling activities of dMi-2 missense mutants. We will first describe the effects of missense mutations mapping to the ATPase domain, then mutations that flank the ATPase domain on the C-terminal side and finally mutations within the PHD finger and chromodomains.

We analysed three point mutations in the ATPase domain: L912V which is located in core 1 and H1151R and R1162Q which are located in core 2 (Fig. 1, Supplementary Fig. 1A). R1162 is the most frequently mutated CHD4 residue in endometrial carcinoma (Fig. 1). This residue is part of the arginine finger motif (ATPase motif VI) which is important for ATP hydrolysis[33]. Unlike R1162, L912 and H1151 do not fall within an established sequence motif and the consequences of their mutation are difficult to predict.

We performed ATPase assays in the absence or presence of saturating amounts of salt dialysis-assembled polynucleosomes (Fig. 2a). All three mutants were capable of hydrolysing ATP in a nucleosome-dependent manner. However, their ATPase activity was reduced compared to the WT protein. The ATPase activities of the L912V mutant and the arginine finger mutant (R1162Q) were reduced to 9% and 19%, respectively. The H1151R mutation had a comparatively minor effect and the mutated enzyme retained 55% of WT ATPase activity.

We then analysed the binding of WT and mutant dMi-2 proteins to a mononucleosome in an electrophoretic mobility shift assay (Fig. 2b). We used a mononucleosome that contains a histone octamer bound to a strong positioning sequence (the Widom 601 sequence) and 80 base pairs of free DNA extending from one side of the histone octamer (0–80 mononucleosome). All dMi-2 proteins formed complexes with the nucleosome that retarded its migration through the gel. We did not detect significant differences in nucleosome binding between WT and mutant dMi-2 proteins.

Next, we assessed nucleosome remodelling activity by a restriction enzyme accessibility (REA) assay (Fig. 2c). This assay makes use of end-labelled DNA fragments which have a nucleosome positioned over an MfeI restriction site (0–80 mononucleosome). This makes nucleosomal DNA refractory to restriction enzyme digestion. Remodelling enzyme-catalysed sliding of the histone octamer along the DNA fragment or (temporary) removal of DNA from the histone octamer surface allows MfeI digestion and results in the generation of a shorter DNA fragment that can be detected and quantified by gel electrophoresis and autoradiography.

Under our reaction conditions, 40 min of incubation with WT dMi-2 protein and ATP made 20% of nucleosomal DNA accessible for MfeI digestion. By contrast, all three missense mutants were severely compromised in facilitating cutting by MfeI. Incubation with the R1162Q mutant resulted in the exposure of only 3% of nucleosomal DNA. We were unable to detect DNA exposure above background after incubation with both the L912V and the H1151R mutant in this assay.

We then employed an assay that more directly measures the ability of remodellers to reposition nucleosomes by sliding them along DNA. In our assay, a mononucleosome which has 77 base pairs extending from the histone octamer (0–77 mononucleosome) is moved from this end position towards the centre of the DNA fragment. This nucleosome repositioning can be detected by a change in electrophoretic mobility (Fig. 2d). No nucleosome sliding by WT dMi-2 was observed in the absence of ATP. In the presence of ATP, incubation with increasing amounts of dMi-2 generated three discernable nucleosome positions which represent histone octamer movement over different distances from the starting position. Again, all three cancer-derived ATPase mutants displayed a severe reduction in activity. The L912V and R1162Q mutants generated one additional nucleosome position that represents histone octamer movement over a short distance. However, compared to WT eight times more protein was required to achieve this effect. We failed to detect any nucleosome sliding activity of the H1151R mutant in this assay.

We also performed nucleosome sliding assays using a centrally positioned mononucleosome with two 77 bp extensions of free DNA (77–77 nucleosome) (Supplementary Fig. 2). With this nucleosome substrate we failed to detect significant remodelling activity with any of the three mutants.

We conclude that all three cancer-derived point mutations mapping to the ATPase domain are compromised for nucleosome remodelling activity in vitro. The effects we have detected after mutating the arginine finger residue Q1162 agree well with previous data obtained with related SNF2 ATPases and verify the

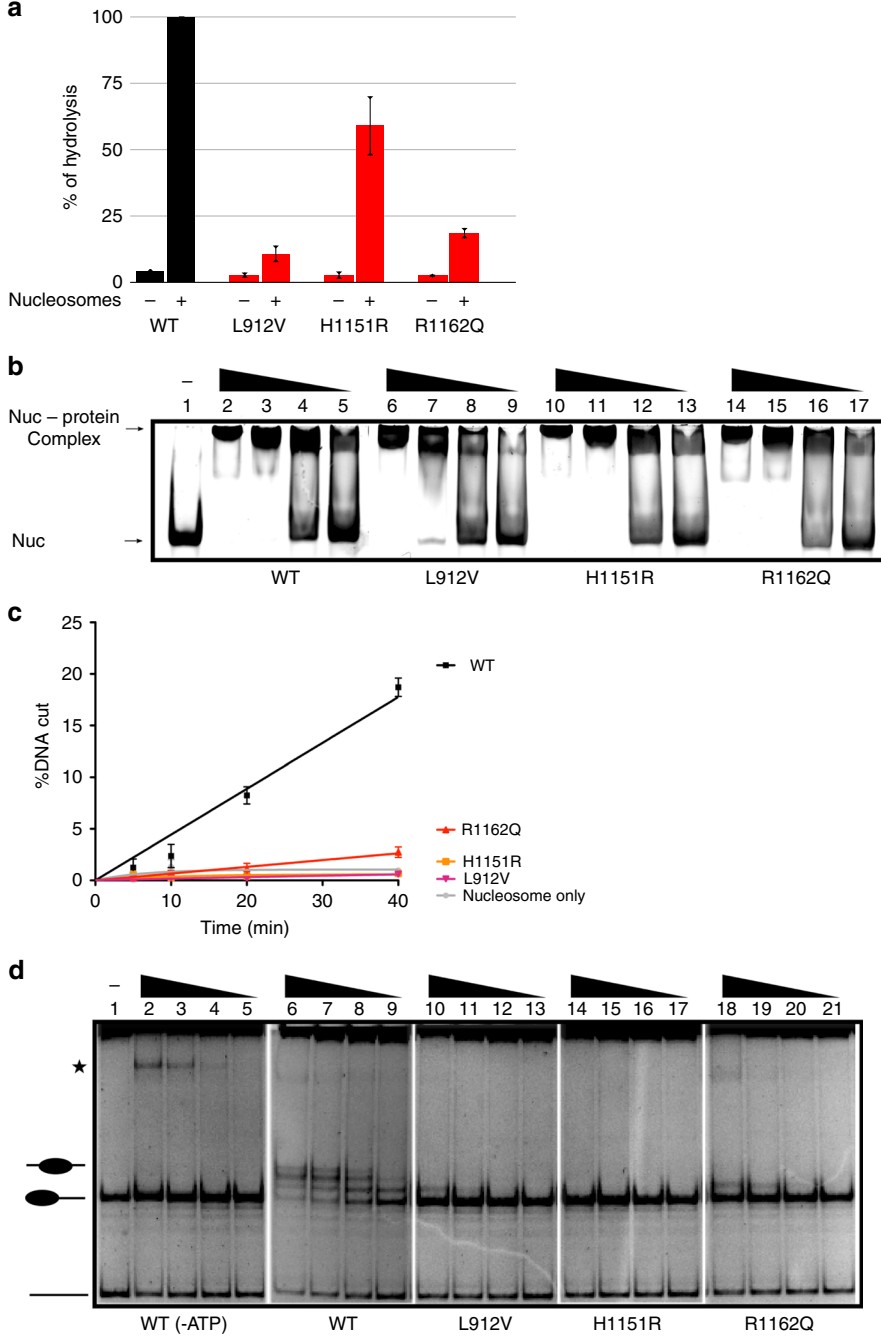

**Fig. 2** ATPase domain mutations disrupt nucleosome remodelling. **a** ATPase activities of wild-type and mutated dMi-2 proteins were determined in absence (−) and presence (+) of saturating amounts of polynucleosomes. ATPase activity of wild-type dMi-2 was set to 100%. Error bars represent SEM and are derived from three independent experiments. **b** Electrophoretic mobility shift assays were carried out with 150 nM of 0–80 mononucleosome and decreasing concentrations of dMi-2 proteins as indicated (lanes 2, 6, 10, 14: 900 mM; lanes 3, 7, 11, 15: 450 nM; lanes 4, 8, 12, 16: 225 nM; lanes 5, 9, 13, 17: 113 nM). The positions of nucleosome–protein complexes and unbound mononucleosome are indicated by arrows on the left. **c** Restriction enzyme accessibility assays were performed with wild-type or mutant dMi-2 proteins (115 nM) and body-labelled 0–80 mononucleosomes (20 nM) as indicated on the right. Reactions were incubated for 5, 10, 20 and 40 min. The percentage of remodelled nucleosomes was determined by dividing the fraction of cut DNA by total DNA at each time point (%DNA cut). Error bars represent SEM and are derived from three independent experiments. **d** Nucleosome sliding assays were carried out with 150 nM of 0–77 mononucleosomes and decreasing concentrations of dMi-2 proteins as indicated (lanes 2, 6, 10, 14, 18: 900 nM; lanes 3, 7, 11, 15, 19: 450 nM; lanes 4, 8, 12, 16, 20: 225 nM; lanes 5, 9, 13, 17, 21: 113 nM). ATP was omitted from reactions shown in lanes 2 to 5 (−ATP). The positions centrally and end positioned mononucleosomes and free DNA are indicated on the left. Asterisk denotes the position of dMi-2/mononucleosome complexes that form at high protein concentrations. All five panels are derived from a single experiment (see Supplementary Fig. 9 for the entire gel). Note that we have reproduced the first two panels (WT and WT-ATP) in Figs. 2d, 3e and 4d to aid visual comparison with the activity of mutants analysed in these figures

importance of the arginine finger motif for efficient ATP hydrolysis and nucleosome remodelling[33]. In addition, we have identified residue L912 of the first ATPase lobe as similarly important for nucleosome-stimulated ATPase activity and nucleosome remodelling. Interestingly, the H1151R mutant had a comparatively mild reduction of nucleosome-stimulated ATPase activity (55% of WT activity) yet displayed the most dramatic loss of remodelling activity. This argues that this

mutation disrupts the coupling of ATP hydrolysis and nucleosome remodelling.

**Mutations on the C-terminal side of the ATPase motor.** In many SF2 ATPases, core 2 is followed by an alpha-helical region called the "brace" (Fig. 3a). The brace is immediately followed by another partially alpha-helical region. In yeast Chd1, this

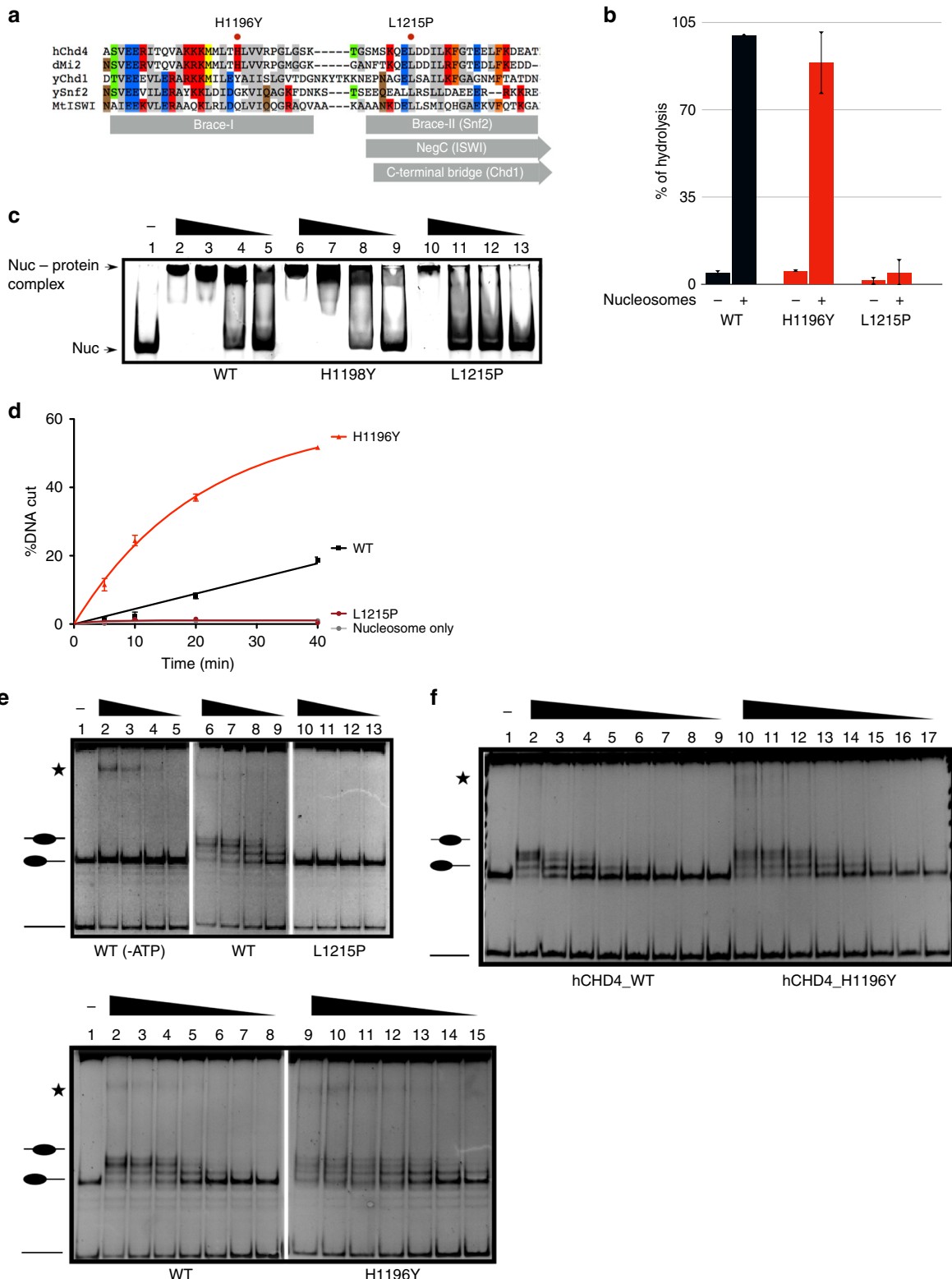

constitutes the "C-terminal bridge", in *Drosophila* ISWI the NegC domain and in yeast Snf2 the Brace-II region. In all three of these remodellers, the brace as well as the bridge/NegC/Brace-II regions are important regulators of the ATPase motor that make contacts with both of its core domains. The amino acid sequence of the brace of Chd1, ISWI and SNF2 show a high degree of conservation with the corresponding regions in CHD4 and dMi-2 suggesting that these also contain a brace helix (Fig. 3a). There is less sequence similarity in the bridge/NegC/Brace-II region but several positions appear to be conserved in CHD4 and dMi-2. One of these conserved residues is L1215 which is mutated in endometrial carcinoma.

We analysed two cancer-derived mutations in these regions: H1196Y which resides in the putative brace of dMi-2 and L1215P which is located within the putative bridge/NegC/Brace-II region. The L1215P mutation decreased ATPase activity to near background levels (Fig. 3b). Consistent with an almost complete loss of ATPase activity we failed to detect nucleosome remodelling activity in REA or nucleosome sliding assays conducted with 0–77 or 77–77 nucleosomes (Fig. 3d, e and Supplementary Fig. 3). dMi-2 L1215P also displayed reduced mononucleosome binding activity suggesting that this region regulates binding of dMi-2 to the nucleosome (Fig. 3c).

By contrast, dMi-2 H1196Y retained full nucleosome-stimulated ATPase (95% of WT activity) and nucleosome binding activities (Fig. 3b, c). Unexpectedly, the H1196Y mutant displayed nucleosome remodelling activities that surpassed those of WT dMi-2 (Fig 3d, e, Supplementary Fig. 3): After 5 min of incubation in an REA assay, when nucleosome remodelling by WT dMi-2 could hardly be detected, the H1196Y mutant had already remodelled 10% of the nucleosomes in the reaction (Fig. 3d). After 40 min, when WT dMi-2 had generated nearly 20% of remodelled nucleosomes, the mutant had remodelled 50% of all nucleosomes. In nucleosome sliding assays remodelling by the H1196Y mutant was observed at four times lower enzyme concentrations compared to WT (Fig. 3e). At higher enzyme concentrations, WT dMi-2 enriched nucleosome positions near the centre of the DNA fragment. By contrast, the H1196Y mutant resulted in a more equal distribution of nucleosome positions across the DNA fragment, indicating that nucleosomes are mobilised towards the centre and towards the ends of the DNA fragment with comparable efficiency. This might suggest that the mutant interprets the underlying DNA sequence differently[34]. In addition, for some remodellers, the direction of nucleosome sliding is determined by the length of the two DNA segments extending from the nucleosome core particle[35]. In order to investigate if dMi-2 H1196Y senses the length of these DNA extensions differently than dMi-2 WT, we tested binding to mononucleosomes with 22 bp (0–22 nucleosome), 44 bp (0–44 nucleosome) and 77 bp (0–77 bp nucleosome) extensions

(Supplementary Fig. 4A). Shortening the DNA segment extending from the core decreased binding affinity for both dMi-2 WT and for dMi-2 H1196Y. However, dMi-2 H1196Y was more strongly affected by this (Supplementary Fig. 4A, compare upper and lower panels). These differences in binding affinity did not translate into differences in ATPase activity. Both dMi-2 WT and dMi-2 H1196Y ATPases were equally stimulated by all three types of mononucleosomes (Supplementary Fig. 4B). Taken together, the results of our remodelling and nucleosome binding assays are consistent with the hypothesis that dMi-2 H1196Y senses DNA extending from the nucleosome core in a different manner. Thus, dMi-2 H1196Y might respond to the DNA sequence of nucleosomal DNA and/or the length of DNA extending from the nucleosome differently resulting in altered nucleosome positioning.

Given the unexpected increase in remodelling activity of the H1196Y mutation, we considered the possibility that this was an inadvertent consequence of analysing *Drosophila* dMi-2 instead of human CHD4 - despite the high degree of sequence conservation between the two proteins. We, therefore, introduced the H1196Y mutation into human CHD4 and compared nucleosome sliding activities of WT CHD4 and CHD4 H1196Y (Fig. 3f). Again, the H1196Y mutation increased nucleosome remodelling activity in this assay.

Taken together, these results suggest regulatory functions for the putative brace and bridge/NegC/Brace-II regions of dMi-2 and CHD4 that are affected by cancer-derived missense mutations.

**Mutations in PHD fingers and chromodomains**. PHD fingers and chromodomains which precede the ATPase domain have also been implicated in regulating catalytic functions of CHD family remodellers[13,23,36]. Three point mutations mapping to the PHD fingers and chromodomains of CHD4 have been identified in endometrial cancer[6,7].

C464Y changes a cysteine residues that binds a zinc ion that is complexed within the PHD finger structure[37]. This mutation is expected to lead to significant structural changes. Introduction of the C464Y mutation did, however, not affect nucleosome binding by dMi-2 (Fig. 4b). dMi-2 C464Y was two- to three-fold less active than WT dMi-2 in ATPase, REA and nucleosome sliding assays but overall retained significant nucleosome remodelling capacity (Fig. 4a, c, d and Supplementary Fig. 5). Since the decreases in ATPase and remodelling activities are of similar magnitude, it is likely that the reduced nucleosome remodelling activity is a direct result of reduced ATPase activity.

Two missense mutations, V558F and R572Y, map to the C-terminal end of the first chromodomain. The crystal structure of yeast Chd1 has revealed that the double chromodomains sit atop the two ATPase cores which adopt an open, catalytically less

**Fig. 3** Mutations in brace and post brace regions alter remodelling activity. **a** Alignment of brace and post brace sequences of yeast Chd1 (yChd1), Snf2 (ySnf2), *M. thermophila* ISWI human CHD4 (hCHD4) and *Drosophila* Mi-2 (dMi-2). Structural elements are indicated. Positions of H1196Y and L1215P mutations are shown. **b** ATPase activities in absence (−) and presence (+) of polynucleosomes. Wild-type dMi-2 activity was set to 100%. Error bars represent SEM from three independent experiments. **c** Electrophoretic mobility shift assays with 150 nM 0–80 mononucleosome and decreasing dMi-2 concentrations (lanes 2, 6, 10: 900 mM; lanes 3, 7, 11: 450 nM; lanes 4, 8, 12: 225 nM; lanes 5, 9, 13: 113 nM). Positions of nucleosome–protein complexes and mononucleosome are indicated. **d** Restriction enzyme accessibility assays were performed with dMi-2 proteins (115 nM) and body-labelled 0–80 mononucleosomes (20 nM) for 5, 10, 20 and 40 min. Percentage of remodelled nucleosomes is shown (%DNA cut). Error bars represent SEM from three independent experiments. **e** Nucleosome sliding assays with 0–77 mononucleosomes (150 nM) and decreasing dMi-2 concentrations. Upper panel: lanes 2, 6, 10: 900 nM; lanes 3, 7, 11: 450 nM; lanes 4, 8, 12: 225 nM; lanes 5, 9, 13: 113 nM. Lower panel: lanes 2, 9: 900 nM; lanes 3, 10: 450 nM, lanes 4, 11: 225 nM; lanes 5, 12: 113 nM, lanes 6, 13: 56.5 nM; lanes 7, 14: 28.25 nM; lanes 8, 15: 14.12 nM. Positions of mononucleosomes and free DNA are indicated. Asterisk: dMi-2/mononucleosome complexes forming at high protein concentrations. Both panels are derived from different experiments. Upper panel is part of gel shown in Supplementary Fig. 9. First two panels are reproduced (WT and WT-ATP) in Figs. 2d, 3e and 4d to aid visual comparison with activity of mutants. **f** Nucleosome sliding assays with 0–77 mononucleosomes (150 nM) and decreasing hCHD4 concentrations. Lanes 2, 10: 900 nM; lanes 3, 11: 450 nM, lanes 4, 12: 225 nM; lanes 5, 13: 113 nM; lanes 6, 14: 56.5 nM; lanes 7, 15: 28.25 nM; lanes 8, 16: 14.12 nM, lanes 9, 17: 7.06 nM

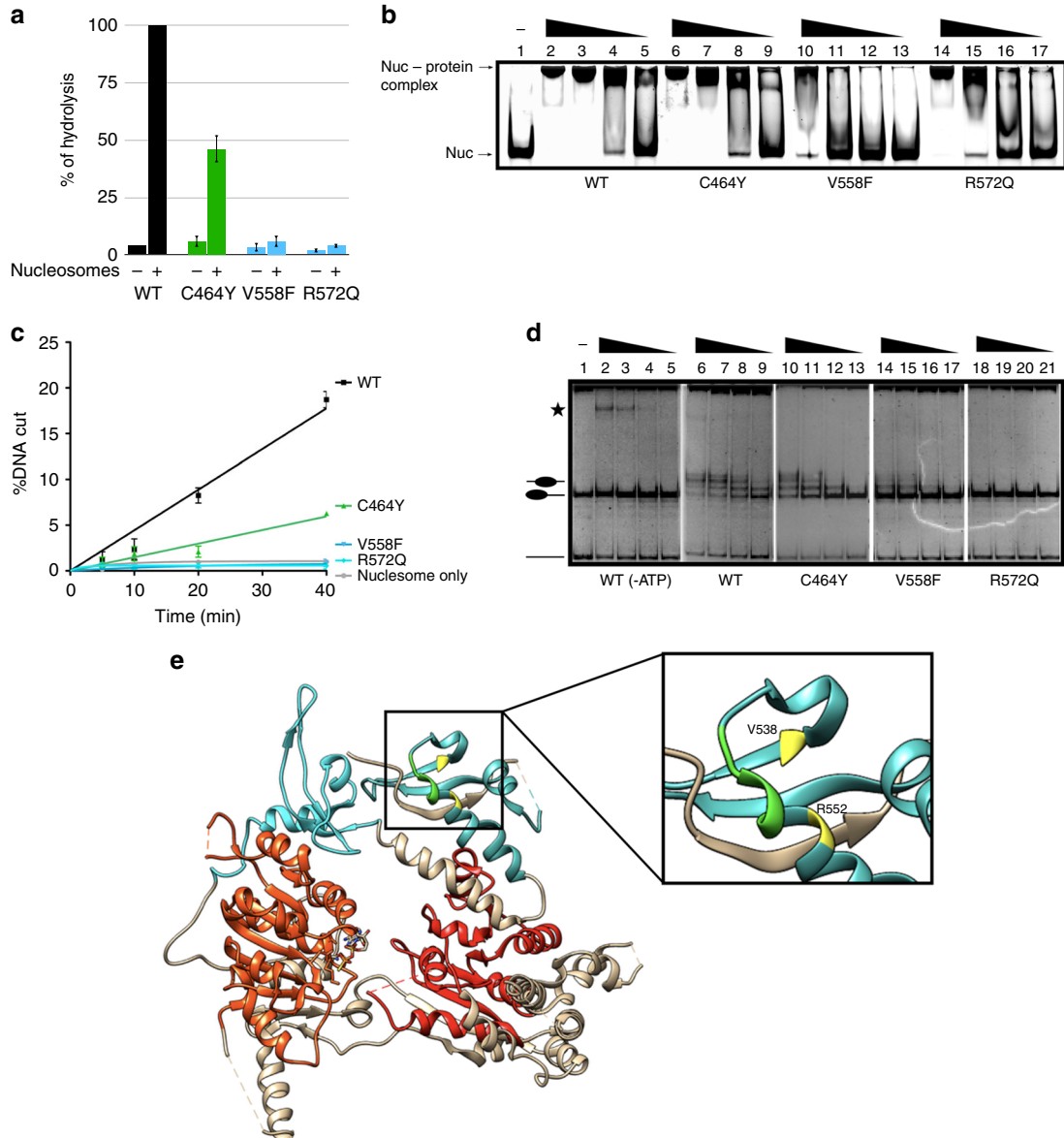

**Fig. 4** Chromodomains mutations abrogate nucleosome remodelling. **a** ATPase activities of wild-type and mutated dMi-2 proteins in absence (−) and presence (+) of polynucleosomes. ATPase activity of wild-type dMi-2 was set to 100%. Error bars represent SEM and are derived from three independent experiments. **b** Electrophoretic mobility shift assays with 150 nM of 0–80 mononucleosome and decreasing dMi-2 concentrations (lanes 2, 6, 10, 14: 900 mM; lanes 3, 7, 11, 15: 450 nM; lanes 4, 8, 12, 16: 225 nM; lanes 5, 9, 13, 17: 113 nM). Positions of nucleosome–protein complexes and mononucleosome are indicated. **c** Restriction enzyme accessibility assays with wild-type or mutant dMi-2 proteins (115 nM) and body-labelled 0–80 mononucleosomes (20 nM). Reactions were incubated for 5, 10, 20 and 40 min. Percentage of remodelled nucleosomes is shown (%DNA cut). Error bars represent SEM from three independent experiments. **d** Nucleosome sliding assays with 0–77 mononucleosomes (150 nM) and decreasing dMi-2 concentrations (lanes 2, 6, 10, 14, 18: 900 nM; lanes 3, 7, 11, 15, 19: 450 nM; lanes 4, 8, 12, 16, 20: 225 nM; lanes 5, 9, 13, 17, 21: 113 nM). ATP was omitted from reactions shown in lanes 2 to 5 (−ATP). Positions of mononucleosomes and free DNA are indicated on the left. Asterisk: dMi-2/mononucleosome complexes forming at high protein concentrations. All panels are derived from a single experiment (shown in Supplementary Fig. 9) except C464Y. The first two panels (WT and WT-ATP) are reproduced in Figs. 2d, 3e and 4d to aid visual comparison with the activity of mutants. **e** Structure of the yCHD1 double chromodomain–ATPase domain fragment (3MWY) (Hauk/Bowman). Inset shows the basic loop of chromodomain 1 of yChd1 (green). The superimposed relative positions of V558 and R572 of CHD4 are shown in yellow. Chromodomain 1: light blue, chromodomain 2: cyan, ATPase domain core 1: orange, ATPase domain core 2: red

active conformation[13]. A negatively charged alpha helix (the chromo wedge) makes direct contacts with core 1 and core 2 residues and restricts access of DNA to the ATPase motor. In this structure, the Chd1 residues corresponding to CHD4 V558 and R572 are located at the opposite side of the chromodomains, facing away from the ATPase domain (Fig. 4e). This suggested that they might not be critical for nucleosome remodelling. Surprisingly, however, both mutants suffered severe reductions of

ATPase, REA and nucleosome sliding activities (Fig. 4a, c, d, Supplementary Fig. 5).

Recently, Bowman and coworkers identified a short basic loop in the first chromodomain of Chd1 that crosslinks to nucleosomal DNA[14]. This contact was also detected in the recent cryo-EM structure of Chd1 bound to the nucleosome solved by the Cramer lab[38]. Superimposition of the dMi-2 amino acid sequence onto the Chd1 crystal structure revealed that V558 and

R572 are in close vicinity of this loop (Fig. 4e). This raises the possibility that mutation of these residues affects binding of dMi-2 to nucleosomal DNA. Indeed, although both chromodomain mutants were clearly able to bind to mononucleosomes in a bandshift assay they did so with reduced affinity (Fig. 4b). This result supports the hypothesis that a region encompassing V558 and R572 makes contacts with nucleosomal DNA that are critical for efficient ATP hydrolysis and nucleosome remodelling.

**Expression of dMi-2 mutants disrupts wing differentiation**. Endometrial cancer arises when epithelial cells lining the uterus leave their differentiation programme and begin to over-proliferate. In order to study the effects of dMi-2 mutants on epithelial differentiation, we chose the *Drosophila* wing. Wing vein formation is a well-established model of epithelial morphogenesis[39].

We established six fly lines carrying UAS-dMi-2 transgenes at the same position in the genome, thereby ensuring similar expression levels[40]. The lines harboured transgenes encoding dMi-2 proteins that were either remodelling competent (WT dMi-2, dMi-2 C464Y, dMi-2 H1196Y) or remodelling defective (dMi-2 H1151R, dMi-2 R1162Q, dMi-2 L1215P). Crossing these transgenic lines with appropriate GAL4 drivers allowed ectopic expression of dMi-2 in the developing fly. First, we used the *engrailed*-GAL4 driver to direct expression of ectopic WT dMi-2 to the posterior part of the developing wing disc. Ectopic posterior crossveins (PCVs) were detected in 48% of wings examined (Fig. 5a, c). We observed branching of the PCV as well as ectopic disconnected veins. This demonstrates that increased dMi-2 activity is sufficient to disrupt the regular PCV differentiation programme and results in aberrant differentiation events. Ectopic expression of the PHD finger mutant dMi-2 C464Y which has mildly reduced remodelling activity in vitro (Fig. 4c, d and Supplementary Fig. 5) produced the same ectopic PCV phenotype but with lower penetrance (28%) (Fig. 5c). Expression of dMi-2 H1196Y which displayed increased remodelling activity in vitro (Fig. 3d, e) produced ectopic PCV phenotypes with increased penetrance (88%) (Fig. 5c).

We classified ectopic PCV phenotypes into two groups: a mild ectopic PCV phenotype (additional, disconnected "dot" of PCV cells or one additional PCV branch) and a strong ectopic PCV phenotype (two additional PCV branches, generation of a "PCV delta"). In flies expressing WT dMi-2 and dMi-2 C464Y, the mild ectopic PCV phenotypes predominated and the strong phenotypes were rarely detected (3% and 0% in flies expressing WT dMi-2 and dMi-2 C464Y, respectively, Fig. 5c). By contrast, in dMi-2 H1196Y expressing flies the strong ectopic PCV phenotypes were more prominent (21%). Thus, both penetrance and severity of ectopic PCV phenotypes correlated remarkably well with the nucleosome remodelling activity of the three dMi-2 proteins tested. This suggests that PCV differentiation is sensitive to an increase in dMi-2 remodelling activity in a dose-dependent manner.

We next analysed expression of three remodelling defective dMi-2 proteins (dMi-2 H1151R, dMi-2 R1162Q, dMi-2 L1215P) in the developing wing. Surprisingly, in all three cases, we again observed a large percentage of wings with ectopic PCV phenotypes (Fig. 5c). However, these crosses also produced partial loss of PCV phenotypes which we had not observed to any significant extent after expression of remodelling competent dMi-2 proteins: Affected PCVs failed to connect to either L4, L5 or both (Fig. 5b). In addition to ectopic PCV and loss-of-PCV phenotypes, we also observed a small number of wings displaying mixed phenotypes, e.g. an additional, ectopic branch in the middle of the PCV combined with a loss of connection to the L4 or L5 vein. The observation that expression of remodelling defective dMi-2 produced loss-of-PCV phenotypes not observed upon expression of remodelling competent dMi-2 suggests that the defective dMi-2 proteins exert a dominant negative or gain of function effect.

The genetic analysis described above was carried out at the normal temperature of 26 °C, which affords optimal GAL4 activity[41]. GAL4 activity is temperature dependent and minimal at 18 °C resulting in much lower transgene expression levels[41]. To test if the observed phenotypes are robust over a range of transgene expression levels, we repeated all genetic crosses at 18 °C (Supplementary Fig. 6). While the penetrance of phenotypes was generally reduced, their distribution was unchanged: ectopic PCV phenotpyes were observed with all transgenes whereas loss-of-PCV phenotypes were restricted to transgenes encoding remodelling defective dMi-2 mutants.

PCV differentiation is very sensitive to changes in BMP/TGFbeta signalling. In order to assess if the expression of genes encoding components of these signalling pathways were derepressed by ectopic expression of dMi-2, we measured the RNA levels in developing wing imaginal discs. We prepared RNA from wing discs from control larvae that carry a UAS-dMi-2 WT transgene but do not express GAL4 (*UAS-dMi-2 wt*) and larvae overexpressing dMi-2 WT (*en-GAL4»UAS-dMi-2 wt*), the hyperactive remodeller dMi-2 H1196Y (*en-GAL4»UAS-dMi-2 H1196Y*) or the remodelling-defective dMi-2 R1162Q mutant (*en-GAL4»UAS-dMi-2 R1162Q*). We then performed RT-qPCR analysis (Supplementary Fig. 7). Measuring dMi-2 mRNA levels confirmed overexpression of the three dMi-2 transgenes. We then tested expression of the BMP/TGFbeta genes *brk, Mad, Med, bi, cv, gbb, dpp* and *tkv*. Two of these genes (*brk* and *Med*) did not show a significant upregulation of RNA levels in wing discs expressing dMi-2 transgenes. The six remaining genes were upregulated between 1.25- and 2-fold with *tkv* showing the strongest effect. While these expression changes are mild, it is worth noting that while RNA was prepared from whole wing imaginal discs the *engrailed* promotor drives expression of dMi-2 transgenes in the posterior half of the wing disc only, suggesting that the observed expression changes are an underestimation. Interestingly, genes encoding BMP/TGFbeta components were similarly upregulated by expression of WT, hyperactive and remodelling-defective dMi-2 proteins. This correlates with the observation that all three types of dMi-2 proteins generate gain of PCV phenotypes and indicates that this phenotype may be caused by expression of ectopic dMi-2 irrespective of its remodelling properties.

Taken together, our results demonstrate that expression of cancer-derived dMi-2 point mutants in the background of endogenous, WT dMi-2 protein upregulates genes encoding BMP/TGFbeta components and is sufficient for derailing the differentiation of epithelial structures.

## Discussion

In this study, we have directly assessed the effects of cancer-associated point mutations in CHD4 on nucleosome remodelling activity. All mutations analysed in this study have been predicted to result in a loss of remodelling activity based on their location within functional domains and the conservation of mutated residues within the CHD4 protein family[6, 7]. However, our study has revealed that these mutations do not always result in an inactive enzyme. Rather, mutations affect ATPase and remodelling activities in a selective and graded manner, resulting in enzymes with diverse nucleosome remodelling potentials: we have identified one mutations that does not disrupt ATPase and

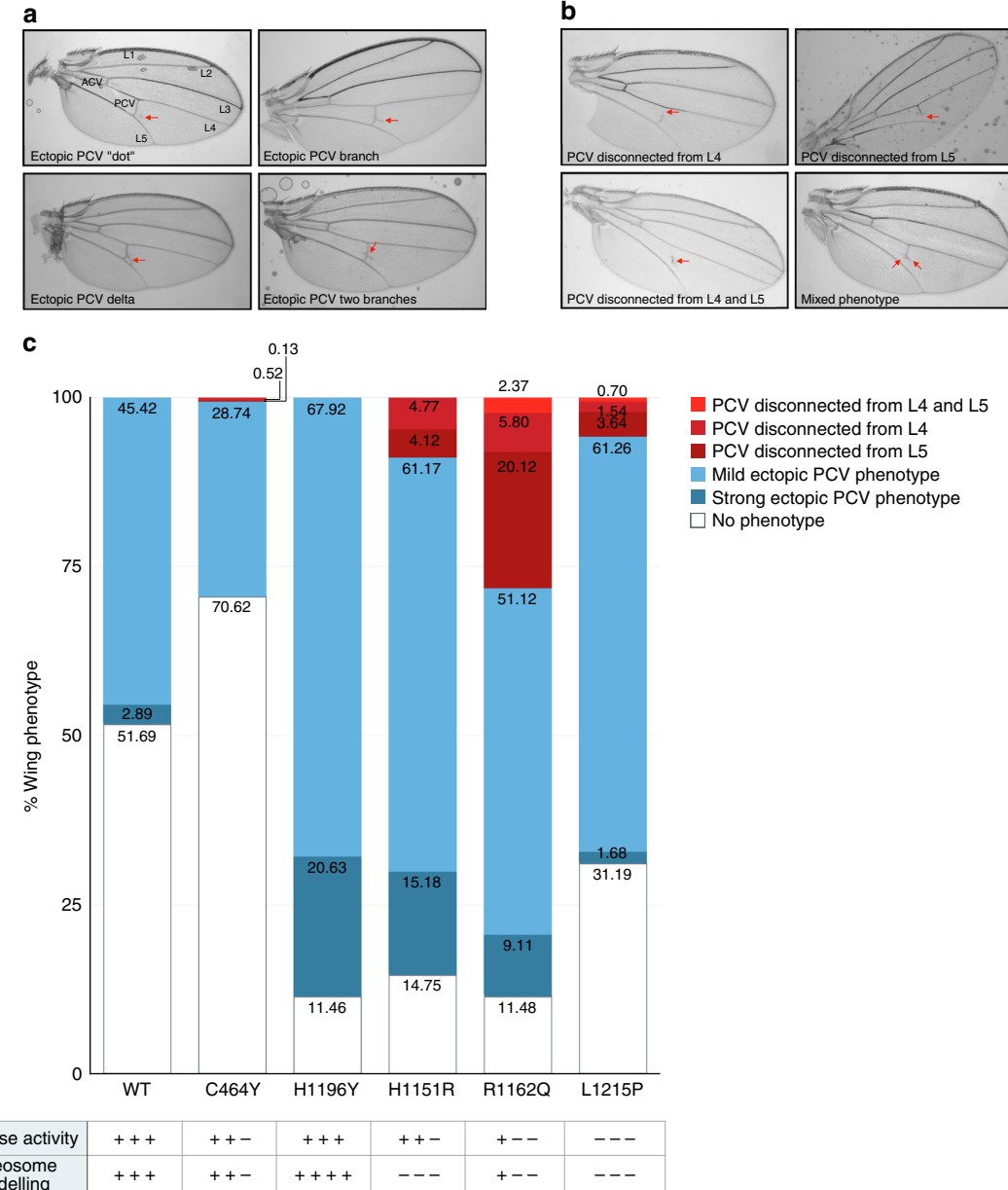

**Fig. 5** Expression of dMi-2 mutants disrupts wing vein formation. **a** Examples of ectopic PCV phenotypes. Mild phenotypes are characterised by formation of an additional disconnected "dot" of PCV cells (upper panel, left) or an ectopic PCV branch (upper panel, right). Strong phenotypes are characterised by the generation of PCV delta (lower panel, left) or two ectopic PCV branches (lower panel, right). Malformed PCV structures are indicated by red arrows. The positions of longitudinal veins (L1 to L5), the anterior (ACV) and posterior (PCV) crossveins are indicated in the first panel. **b** Examples of loss of PCV phenotypes. Mild phenotypes are characterised by disconnection of the PCV from L4 or L5 (upper panels, left and right). Strong phenotypes are characterised by loss of connection from both, L4 and L5 (lower panel, left). A small percentage of wings showed a mixed phenotype with both ectopic PCV structures and loss of PCV structures (lower panel, right). **c** Graph showing the distribution (in %) of ectopic PCV (blue) and loss of PCV (red) phenotypes for expression of wild-type and mutant dMi-2 proteins at 26 °C as indicated. Relative ATPase and nucleosome remodelling activities for each dMi-2 protein as determined in Figs. 2–4 are shown at the bottom

remodelling function but results in a comparatively mild reduction of these activities. By contrast, other mutations strongly decrease ATPase and remodelling activities to levels below the detection limit of our assays, whereas one mutant shows significantly increased remodelling capacity. Remodelling by this mutant also produced an altered nucleosome positioning pattern. Moreover, we identified two mutants where the efficiency of coupling of ATP hydrolysis and remodelling is changed. This diversity of outcomes was unexpected and sheds new light on the molecular mechanisms disturbed in cancer.

The dMi-2 H1151R mutant retains robust ATPase activity but we failed to detect nucleosome remodelling in two independent assays. By contrast, other mutants with ATPase activities below that of dMi-2 H1151R did remodel nucleosomes (e.g. C464Y). This argues that the reduced ATPase activity of H1151R is not sufficient to explain its severe loss of remodelling activity. Rather, these results suggest that the H1151R mutant is not able to efficiently couple ATP hydrolysis to nucleosome remodelling. H1151 is located between ATPase motifs V and VI (Supplementary Fig. 1A). Whereas ATPase motif V, which is essential for proper

coupling of ATP hydrolysis and nucleosome remodelling, is contacting DNA in many SF2 ATPases, ATPase motif VI, together with ATPase motifs I and II, coordinates binding and hydrolysis of ATP[33,42]. Its position suggests that H1151 is important for the communication between ATP and nucleic acid binding regions of dMi-2 and for the allosteric changes that are prompted by the hydrolysis of ATP.

Coupling of ATPase and remodelling was also altered in dMi-2 H1196Y. H1196 is located on the C-terminal side of the ATPase domain in a region analogous to the brace in other remodellers. The brace contacts the two ATPase cores and contributes to keeping them in a conformation that is not conducive to ATP hydrolysis. These brace-core contacts are released when the remodeller engages nucleic acids favouring ATP hydrolysis and nucleosome remodelling. The inhibitory function of the brace region is maintained in the H1196Y mutant as it still is dependent on nucleosomes for efficient ATP-hydrolysis. The ATPase activity of dMi-2 H1196Y is similar to that of WT dMi-2. However, dMi-2 H1196Y has strongly increased nucleosome remodelling activity. This argues that dMi-2 H1196Y couples ATP hydrolysis and nucleosome remodelling more efficiently. In addition, this mutant also altered the distribution of nucleosome positions generated on our test templates. The mechanistic basis for this is presently unclear.

In the past, coupling mutants of several nucleosome remodellers have been generated by mutagenesis to analyse the molecular mechanisms of nucleosome remodelling[17,19,22,43]. To our knowledge, CHD4 H1151R and CHD4 H1196Y are the first coupling mutants identified in cancer cells. This demonstrates that the coupling of ATP hydrolysis and nucleosome remodelling is, indeed, targeted in cancer cells.

Recently, protein domains C-terminal to the brace have been demonstrated to help regulate the activity of the ATPase motor. These regulatory regions include the C-terminal bridge of Chd1, the NegC domain of ISWI and the Brace-II alpha-helix of Snf2. All three of these regions are characterised by a high alpha helix content. No structural information on the corresponding regions of CHD4 or dMi-2 is available. However, this region is predicted to form alpha-helices and several residues are conserved between CHD4, dMi-2, Chd1, ISWI and Snf2 (Fig. 3a). In endometrial cancer, the conserved residue L1215 within this region is mutated to a proline residue which would disrupt an alpha-helix. Indeed, the L1215P mutation results in a severe loss of ATPase and nucleosome remodelling activity. It is conceivable that the mutation disrupts intramolecular interactions with the ATPase motor that are critical for efficient remodelling. Alternatively, but not mutually exclusively, the affected region might contribute to substrate binding by mediating contacts to the nucleosome. In agreement with the latter scenario, we have observed a reduction of nucleosome binding by the dMi-2 L1215P mutant. Although the mechanistic details remain to be resolved, our analysis clearly shows that the region adjacent to the brace is critical for CHD4/dMi-2-mediated nucleosome remodelling. This identifies a novel regulatory region in CHD4 enzymes.

Mutation of the zinc binding C464 resulted in a mild reduction in dMi-2 ATPase and nucleosome remodelling activity. Our results agree well with previous studies of the role of PHD fingers in human CHD4[36]: Deletion of both PHD fingers from recombinant CHD4 results in a minor decrease of ATPase and nucleosome sliding activities. Moreover, the decrease in ATPase activity is only apparent with unmodified nucleosomes but not with nucleosomes containing methylated H3K4. This is consistent with H3K4 methylation lowering the affinity of CHD4 PHD finger binding[36]. The nucleosomes we have used in this study were assembled with histone octamers purified from Drosophila embryos. These contain a mixture of unmodified and H3K4 methylated nucleosomes.

Taken together, these results suggest that disruption of the PHD finger structure only mildly affects the nucleosome remodelling potential of CHD4/dMi-2. It is possible that the PHD fingers play more important roles for other aspects of CHD4 function in vivo, such as interaction with subunits of the NuRD complex or stable association with chromatin. However, we note that expression of dMi-2 C464Y in the developing wing also has a comparatively weak effect.

In contrast to the mutation in the PHD fingers, two cancer-derived mutations mapping to the first chromodomain strongly impair ATPase and remodelling activity of dMi-2. This corroborates results of a previous deletion analysis showing that the double chromodomain region is essential for nucleosome remodelling[44].

Why do single point mutations in the chromodomains have such a profound effect? Recently, the structure of Chd1 bound to the nucleosome has been analysed by cryo EM and crosslinking studies[14,16,38]. These structural studies suggest that when Chd1 engages the nucleosome, the chromodomains rotate away from the ATPase domain and bind the nucleosome. A short basic loop at the end of chromodomain 1 has been proposed to directly contact nucleosomal DNA at SHL1[14]. This movement of the chromodomains away from the ATPase domain is expected to disrupt inhibitory chromo wedge-ATPase motor interactions, thereby facilitating ATP hydrolysis and nucleosome remodelling.

Superimposition of the dMi-2 sequence onto the Chd1 structure reveals that V558 and R572 are in the vicinity of this short basic loop (Fig. 4e). Previous sequence alignments have indicated that the Chd1 short basic loop is conserved in CHD1, CHD2 and CHD6-9 but not in CHD4[14]. However, visual inspection of the surrounding sequences show that CHD4 and dMi-2 do contain a basic region that appears to be shifted with respect to the basic loop of Chd1 (Supplementary Fig. 8). The R572Q mutation maps to this basic region. We propose that V558 and R572 are part of a structural element that, like the basic loop of the Chd1 chromodomains, contacts nucleosomal DNA during the remodelling reaction. In agreement with this hypothesis, we have shown that while dMi-2 V558F and dMi-2 R572Y still bind to the nucleosome, they do so with reduced affinity. The almost complete loss of nucleosome remodelling activity of both chromodomain mutants provides strong support for the structural models proposed by the Bowman and Cramer labs and demonstrates the functional importance of this chromodomain–nucleosome interaction.

It is clear that the majority of CHD4 mutations identified in endometrial cancers negatively impact ATPase and/or remodelling activity. It is conceivable that a reduction of overall CHD4 activity in endometrial cancer cells contributes to cancerogenesis. However, CHD4 gene deletions or frameshift mutations—which would likewise lower overall CHD4 activity—are very rare in endometrial cancer[6]. By contrast, missense mutations predominate (89%) arguing that it might be the mutated CHD4 protein itself that contributes to cancerogenesis via gain-of-function or dominant negative mechanisms. This hypothesis is supported by our observation that expression of mutated dMi-2 transgenes is sufficient to misregulate wing differentiation even though these flies possess two WT copies of the endogenous dMi-2 gene. It is conceivable that CHD4 mutations result in altered nucleosome positioning in vivo, making regulatory sequences that govern gene transcription more or less accessible. This might contribute to the activation of oncogenes or the repression of tumour suppressor genes (TSGs). Indeed, CHD4 has recently been shown to extend the repression of TSGs during DNA damage. Importantly, the ATPase function of CHD4 is required for this process[45]. Of the mutations tested in this study, only one (H1196Y) changed the distribution of nucleosome positions

generated with our test templates in vitro. However, it is important to stress that these templates contain an artificial, strong positioning sequence and might not reveal more subtle differences in nucleosome positioning.

The effects of CHD4 mutations might also manifest themselves by an altered activity of the NuRD complex. NuRD combines CHD4 nucleosome remodelling with HDAC1/2 histone deacetylase activities. Early work has suggested that remodelling is a pre-requisite for efficient nucleosome deacetylation[2]. Thus, altered CHD4 activity is likely to have an impact not only on nucleosome positioning but also on histone acetylation.

Endometrial cancer arises when epithelial cells of the uterus become transformed and overproliferate. In serous endometrial carcinoma, the most aggressive from of this cancer, CHD4 mutations are most frequent and occur alongside mutations in tumour suppressor or oncogenes including TP53, PIC3CA, PPP2R1A or FBXW7[6]. The individual contribution of CHD4 mutations to malignant transformation is, therefore, difficult to assess. We have used a classical model for epithelial morphogenesis, the *Drosophila* wing, to analyse the effects of expression of dMi-2 mutants. While we did not observe an overproliferation of epithelial cells, our results demonstrate that expression of dMi-2 is sufficient to derail the PCV differentiation programme. PCV phenotypes that are similar to those we have observed upon dMi-2 expression have been reported in flies with gain- or loss-of-function of other chromatin regulators, including snr1, COMPASS and LSD1[46–48]. These and the findings reported in this study suggest that PCV differentiation is very sensitive to epigenetic imbalances.

PCV differentiation is under control of TGF-beta/BMP signalling[49]. Indeed, we have observed increased expression of genes encoding TGF-beta/BMP signalling components in wing discs expressing ectopic dMi-2 mutants which may contribute to altered TGF-beta/BMP signalling or its readout. In mammals, there are several examples for a role of CHD4 in TGF-beta/BMP signalling: the CHD4 containing NuRD complex interacts with TGF-beta/BMP signalling factors and regulates TGF-beta/BMP target genes during cellular differentiation in different contexts[50–53]. It is conceivable that altered nucleosome positioning at TGF-beta/BMP target genes changes their transcription and consequently wing cell fate.

We note that TGF-beta/BMP signalling is not only crucial for PCV differentiation in the developing fly wing but that it also plays an important role in endometrial cancer initiation and progression[54,55]. It is, thus, possible that the molecular mechanisms used by mutant dMi-2 proteins to derail PCV differentiation are similar to the mechanisms used by mutant CHD4 proteins to contribute to transformation of epithelial cells of the endometrium. A role of the CHD4-containing NuRD complex in the transition of a mesenchymal cell type into an epithelial cell type in the context of oral squamous cell carcinoma cell lines has recently been demonstrated, lending further support to the idea that CHD4 function is critical for epithelial cell fate[56].

Our detailed mechanistic analysis of cancer-derived dMi-2/CHD4 mutants has revealed defects in different structural domains resulting in diverse effects including lower ATPase activity, decreased nucleosome binding, inefficient coupling of ATPase and remodelling activities and altered nucleosome positioning. This heterogeneity of defects and consequences implies that future epigenetic drugs aimed at restoring the activity of mutant CHD4 will have to be specific for the patient and the CHD4 mutation in question. Do different CHD4 mutants create different epigenetic landscapes in cancer cells and if so, would patients then benefit from treatment with different epigenetic drugs? Our work provides a basis for addressing these questions in the future.

## Methods

**Generation of baculoviruses and cell culture**. FLAG tagged dMi-2 WT (AF119716.1) was cloned to pFastBacDual vector using NotI and XbaI. All mutants were generated by site-directed mutagenesis using appropriate sets of primers (Agilent Technologies Mutagenesis Kit). Constructs were verified by DNA sequencing. pFBD-dMi-2 WT and mutants were transformed into DH10Bac *Escherichia coli* for generation of recombinant bacmid. Bacmids were transfected into Sf9 cells for generation of baculoviruses using the Cellfectin II reagent (Invitrogen). Baculoviruses were amplified twice before they were used for infection of cells for protein production. Cells were harvested 72 h after infection. Sf9 cell lines were maintained at 26 °C in Sf-900 medium (Gibco), supplemented with 10% FBS (foetal bovine serum) and 1% penicillin/streptavidin (Gibco). Sequences of primers used for site-directed mutagenesis:

dMi-2_C452Y_fwd: TTGCTGTGCTACGACTCATGTCCCTCC, dMi-2_C452Y_rev: GGAGGGACATGAGTCGTAGCACAGCAA, dMi-2_V538F_fwd: TGCGAATGGTTCCCCGAAGTGCAACTGGAC, dMi-2_V538F_rev: GTCCAGT TGCACTTCGGGGAACCATTCGCA, dMi-2_R552Q_fwd: CTCATGATTCAG TCGTTCCAGCGCAAG, dMi-2_R552Q_rev: CTTGCGCTGGAACGACTGAA TCATGAG, dMi-2_L914V_fwd: CTCGAGGAGGTGTTCCATCTGCTCAACT TC, dMi-2_L914V_rev: GAAGTTGAGCAGATGGAACACCTCCTCGAG, dMi-2_H1153R_fwd: AATCCCCGAAACGATATTCAGGCC, dMi-2_H1153R_rev: GGCCTGAATATCGTTTCGGGGATT, dMi-2_R1164Q_fwd: CTTCTCCCGAG CCCATCAGATTGGCCAGGCTAACAA, dMi-2_R1164Q_rev: TTGTTAGCC TGGCCAATCTGATGGGCTCGGGAGAAG, dMi-2_H1198Y_fwd: CGTAAG ATGATGTTGACTTATCTTGTGGTCC, dMi-2_H1198Y_rev: GGACCACAAGA TAAGTCAACATCATCTTACG, dMi-2_L1217P_fwd TTTACAAAG CAAGAA CCGGACGATATCCTTCGT, dMi-2_L1217P_rev ACGAAGGATATCGTCCGG TTCTTGCTTTGTAAA.

**Purification of recombinant proteins**. Sf9 cells were washed 3× in 1× PBS, pelleted by centrifugation and resuspended in Lysis buffer (200 mM KCl, 20 mM HEPES pH 7.6, 0.1% NP-40, 10% glycerol) and subjected to three freeze/thaw cycles in liquid $N_2$[24]. Lysate was centrifuged (30 min, 16,200×*g*, 4 °C, Heraeus Fresco 17) and the supernatant was incubated for 4 h with anti-FLAG agarose beads (Sigma, A2220). Beads were transferred to a column and washed with increasing salt concentrations (200/500/1000 mM KCl, 20 mM HEPES pH 7.6, 0.1% NP-40, 10% glycerol) before elution with excess FLAG peptide (in 20 mM Tris pH 8, 150 mM KCl, 10% glycerol). All buffers were supplemented with protease inhibitors. Purified proteins were separated by SDS-PAGE and visualised with silver or Coomassie staining.

**Histone purification and nucleosome assembly**. *Drosophila* histones were purified from 100 g dechorionized 0–12 h embryos over a hydroxyapatite column[57,58]. Mononucleosomes were assembled on 224 bp (0-NPS-77), 227 bp (0–80) and 301 bp (77-NPS-77) DNA fragments. Polynucleosomes were assembled on the pUC12x601 plasmid containing 12 copies of the 601 nucleosome positioning sequence. DNA and histones were mixed at an approximately 1:1 ratio in high-salt buffer (10 mM Tris, 2 M KCl, 1 mM EDTA, 1 mM, 1 mM β-mercaptoethanol). Continuous dialysis was performed over 24 h at 4 °C by reducing the salt concentration to 50 mM. Polynucleosome assembly was analysed by Not I digestion. Mononucleosome assemblies were resolved over 5% native PAA gel. Mononucleosome template for REA and bandshift experiments (0–80) were separated from overassembled nucleosomes and free DNA by ultracentrifugation (18 h, 217,485×*g*, 4 °C in Beckman Optima LE-80K, rotor SW40Ti). All DNA templates for REA experiments were radioactively labelled with α-$^{32}$P-dCTP (3000 Ci/mmol, 10 mCi/mL, Hartmann Analytic).

Sequences of primers used for generating DNA fragments used for mononucleosome assembly: 601_147_fwd: CTGGAGAATCCCGGTGCC, +80_601rev: TCGGTACCCGGGGATCC, 0–77fwd: GATCCA GAATCCTGGTGCTG AG, 0–77rev: GTACAGAGAGGGAGAGTCACAAAAC, 77–77fwd: ATCTTTT GAGGTCCGGTTCTTT, 77–77rev: GTACAGAGAGGGAGAGTCACAAAAC.

**Nucleosome sliding assay**. Sliding assays were performed on 0-NPS-77 and 77-NPS-77 nucleosomal templates (150 nM). Typical reactions were performed in 10 μL BC100 buffer (20 mM HEPES, 100 mM KCl, 0.4 mM EDTA, 10% glycerol) with varying concentrations of dMi-2 WT or mutants, 2 mM ATP and 6 mM MgCl₂. Reactions were incubated for 5 min at 26 °C before addition of ATP and further incubated for 45 min. Reactions were stopped by addition of competitor plasmid DNA (200 ng/μL) and incubation for 10 min on ice. Reactions were loaded on 5% PAA native gels and run in 0.5×TBE. Gels were stained with SyBr Gold reagent or ethidium bromide. Gels were scanned with a Biorad ChemiDoc Touch Imaging system.

**Nucleosome bandshift assay**. Bandshift assays were performed as the sliding assays described above using 0–80 mononucleosome with several modifications: ATP and competitor DNA were excluded from the reaction. Reactions were incubated for 30 minutes at 26 °C, after which they were loaded on 5% PAA native gels and run in 0.5×TBE at 4 °C. Gels were stained with SyBr Gold reagent. Gels were scanned with Biorad ChemiDoc Touch Imaging system.

**Restriction enzyme accessibility assay (REA)**. The REA was carried out on a [32]P-labelled mononucleosome. In this mononucleosome, the histone octamer occupies a 601-positioning sequence containing a MfeI restriction site[25,59,60]. The nucleosome protects this site from digestion by MfeI. REA reactions in the presence of MfeI were initiated by the addition of ATP mix (2 mm ATP/6 mM $MgCl_2$). Aliquots were removed at various times and quenched in 1.5 volumes of 10% glycerol, 70 mM EDTA, 20 mM Tris (pH 7.7), 2% SDS, 0.2 mg per mL xylene cyanole and bromophenol blue. The samples were deproteinised by proteinase K digestion and DNA fragments were separated on 5% native polyacrylamide gels. Gels were dried and exposed to Phosphoimager screen. Signal quantification was done using the Science Lab Image Gauge (FUJIFILMS) software.

**ATPase assay**. A typical 15 μL ATPase assay reaction was performed in BC buffer (20 mM HEPES, 0.4 mM EDTA, 10% glycerol) in presence of saturating amounts of polynucleosomes, varying concentrations of dMi-2, 20 μM ATP and trace amounts of $(\gamma\text{-}^{32}P)$-ATP (3000 Ci/mmol, 10 mCi/mL, Hartmann Analytic)[23,61]. Protein dilutions were prepared in BC100 buffer (20 mM HEPES, 100 mM KCl, 0.4 mM EDTA, 10% glycerol). Reactions were started by pipetting them into a mixture of unlabelled and labelled ATP. Reaction was continued at 26 °C for 30 min. 1.5 μL of each reaction was spotted on a TLC PEI Cellulose plate (Millipore) and dried at room temperature. Hydrolysed phosphate was separated from unhydrolysed ATP in 0.5 M LiCl/1 M formic acid buffer using thin layer chromatography. Dried plates were exposed to Phosphoimager screen and signals were quantified using Science Lab Image Gauge (FUJIFILM) software.

**ATP filter binding assay**. Typical reactions were performed in a 15-μL volume containing 90 nM dMi-2 WT or mutant, 0.25 mM $MgCl_2$ and 0.0375 μL of $(\gamma\text{-}^{32}P)$-ATP (3000 Ci/mmol, 10 mCi/mL). Binding reactions were performed in BC100 buffer (20 mM HEPES, 100 mM KCl, 0.4 mM EDTA, 10% glycerol) for 5 min at 26 °C before addition of ATP and further incubation for 30 min; 2 μL of the reaction was spotted on nitrocellulose membrane under vacuum and washed with approximately 150 mL of BC100 buffer to wash away unbound ATP. Membranes were air dried and exposed to a Phosphoimager screen. Signal quantification was done using the Science Lab Image Gauge (FUJIFILMS) software.

**Generation of dMi-2 transgenic fly lines and genetic crosses**. PCR amplified dMi-2 WT and mutant fragments were cloned into pUASt-attB-rfa vector (gift from Sven Bogdan), using the pENTR Directional Topo cloning kit (Invitrogen) and the Gateway LR Clonase II Enzyme mix (Invitrogen). Purified vectors were injected into *Drosophila* embryos (y[1] M{vas-int.Dm}ZH-2A w[*]; M{3xP3-RFP. attP}ZH-86Fb) which have a *attP* landing site on the 3rd chromosome (Bischof et al. 2006). The F0 generation was crossed with the w[1118] strain and then between each other, until homozygous flies were obtained. The presence of the transgene was confirmed by PCR. UAS flies were crossed with engrailed GAL4 flies, directing ectopic expression in the posterior part of the developing wing. All flies were collected as virgin flies before setting up the crosses. Flies were kept at 26 °C in a fly incubator and on standard food[62].

**RT-qPCR**. Wing imaginal discs were dissected from late L3 larvae according to standard procedures. In brief, larvae were collected and washed in PBS. Wing imaginal discs were dissected in ice-cold PBS and kept on ice for a maximum of 1 h. After pelleting the tissue, PBS was removed and the samples were snap frozen in liquid nitrogen and kept at −80 °C until RNA isolation. Roughly 100 wing imaginal discs were collected per genotype. RNA isolation was performed using the peqGOLD Total RNA Kit (S-Line, Peqlab) according to the manufacturer's protocol. In total 10–20 μg of RNA was isolated. cDNA was synthesised from 1 μg of isolated RNA using the SensiFAST™ cDNA Synthesis Kit (Bioline) according to the manufacturer's protocol. cDNA was diluted ten times for further use in RT-qPCR reactions. cDNA was synthesised in triplicate. RT-qPCR analysis was performed using the SensiFAST™ SYBR® Lo-ROX Kit (Bioline) on a Mx3000P cycler (Agilent Technologies) according to the instruction manual. Calculations were done according to ref. [63]. Data presented in the graphs represent mean values of three technical replicates with standard deviation.

**Data availability**. All data generated or analysed during this study are included in this published article (and its supplementary information files). Data are available from the authors upon reasonable request.

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

## Acknowledgements

We are grateful to Dr. Ina Theofel, Tim Hundertmark and Ruth Hyland for embryo injections and help with generation and maintenance of fly lines and to Sven Bogdan for the gift of *Drosophila* expression vectors. We thank Karim Bouazoune for comments on the manuscript. K.K., S.A., C.R. and A.B. were supported by DFG grant TRR81/A01, I.M. by DFG grant 2102/6-2, F.F. and R.M. by DFG grant MU 601/17-1, H.H., A.F. and G.L. by DFG grant SFB960/B2.

## Author contributions

K.K., A.S., I.M., S.A., H.H. and A.F. planned and performed the experiments, F.F. contributed to bioinformatic analysis, R.M., C.R., G.L. and A.B. planned and supervised the experiments and wrote the manuscript.

## Additional information

**Competing interests:** The authors declare no competing interests.

