## [Peer Review File · Nature Communications]

Reviewers' Comments:

Reviewer #1:

Remarks to the Author:

ATP-dependent chromatin remodelers, most notably the BAF complex, are frequently mutated in human cancers. The precise links between chromatin remodeling and cancer pathologies are not clear. Here, the authors mutated the *Drosophila* chromatin remodeler dMi-2 at residues that are frequently mutated in human CHD4 in endometrial carcinoma. The authors characterise the nucleosome remodelling properties of these dMi-2 mutants, including their nucleosome binding affinity, ATPase activity and nucleosome remodelling efficiency. Except the H1198 mutation, all other mutations decrease nucleosome-remodeling activities *in vitro*. As in the case of ISWI, the data show that a decrease in remodeling activity does not always reflect a decrease in ATPase activity. The authors report that regions outside the ATPase domain, such as the chromodomains, are essential for the remodeling activity. Interestingly, the mutations point to a region, downstream to the ATPase domain, which can regulate the activity and direction of the remodeler without changing its ATPase activity. Finally, expression of these dMi-2 mutants in flies lead to ectopic branching or disconnected posterior crossveins in fly wings.

Authors have performed careful and comprehensive *in vitro* characterisation of the remodeling activities of the dMi-2 mutants. The data is convincing and important to understand the roles of CHD4 in human cancers. However, it is not clear if the changes in remodeling activities are the sole reason for the wing phenotype or the human cancer pathologies. The authors have already generated flies carrying these mutants, it should be relatively easy to provide more insights on the *in vivo* molecular mechanisms underlying the wing phenotypes. I therefore recommend publication with a revision to strengthen the *in vivo* data in the manuscript.

Major suggestions:

1. I suggest performing RNAseq or real time qPCR experiments in the wing disc or other tissues of the flies carrying these mutants, to test which dMi-2 bound genes are affected. Alternatively, authors can simply use real time qPCR to test TGF-beta/BMP signaling pathway components, as suggested in the manuscript. It will be very interesting if the affected genes turn out to reflect the abnormal gene expression observed in human cancers. It will also be advisable if the authors can map the *in vivo* nucleosome patterns on these affected genes in the mutants to understand the role of the remodeling activities. Provided that expressing the wild type Mi-2 alone already gives a wing phenotype due to overexpression, it may be better to express the wild-type/mutants in a heterozygous Mi-2 null-mutant background. It can imitate the heterozygous CHD4 mutant situations in human cancers and it hopefully also reduce the overall Mi-2 level to close to physiological endogenous level.

2. Authors reported an interesting observation that H1196Y mutation renders the remodeler with the same ATPase activity as the wild type Mi-2, but it becomes a better remodeler with no directionality to move the nucleosome to the center. It may suggest that the brace-I region is responsible for measuring DNA length and give wild-type Mi-2 the directionality. Can the authors test binding between wt/H1196Y Mi-2 and mononucleosome with 10bp, 20bp, 30bp and 60bp overhang? Or test the ATPase activity with different overhang length? It may be that H1196Y can no longer distinguish a short vs a long DNA overhang. The authors can also provide other experiments that may explain the change in directionality of the enzyme.

Minor suggestions:

1. Simply increasing wild type Mi-2 protein amount is enough to cause a phenotype in the wing. This suggests that total protein level is important to interpret the result. The authors may provide western blot showing protein level of exogenous mutated Mi-2 and endogenous Mi-2 in these transgenic flies. The mutants are expressed from the same genomic loci, but the protein stability of different mutants may vary.
2. The title and abstract may cause misunderstanding. Almost the whole manuscript is based on the *Drosophila* protein Mi-2, but the abstract led me one think that the *in vitro* experiments were

done with human proteins and human mutations. *Drosophila* was only mentioned in the genetic experiments. It would be good to make that clearer.

3. It is not clear to me if how L1215P disrupt nucleosome binding. Does the brace II region bind directly to nucleosomal DNA and L1215P disrupt the interaction with the nucleosome, or brace II bind to the ATPase core to inhibit its interaction with the nucleosome, and L1215P enhance the interaction with ATPase. Does the brace II region alone bind to nucleosome? Maybe the authors can elaborate.

4. It is unclear how the mutation in the PHD domain may affect the ATPase activity. In order to understand the function of this mutation, the authors may try to use nucleosomes that are methylated at H3K4 residue to perform their binding assay and ATPase assay. Only a minor fraction of endogenous histone octamer extracted from the cell contains H3K4me3. If it has been reported before in the literature, please elaborate.

5. Figure 1 and 2 could be easily combined.

Reviewer #2:

Remarks to the Author:

Referee's Report for the manuscript entitled "Tumour-associated missense mutations in CHD4 ATPases alter their nucleosome remodelling properties in a patient-specific manner" by Kristina Kovač et. al.

In this paper, the authors systematically characterise CHD4 mutations found in endometrial cancers. The interesting finding is that they identify mutations that increase, as well as decrease, the enzyme's activity. The in-vitro characterisation of the mutants is complemented by in-vivo experiments in *Drosophila*, which show that the expression of cancer-derived CHD4 mutants is sufficient to misregulate differentiation of epithelial structures. I thought this was a carefully designed and well thought out study, which is also clearly presented, and it deserves to be published.

In the Abstract, I think the authors should conclude with "Our results define the defects of CHD4 in cancer...".

In the Introduction, the authors argue that CHD4 mutants do not contribute to disease by lowering CHD4 activity (haploinsufficiency). However, it is not clear to me why the results don't simply suggest that CHD4 levels are critical and that both down-regulating and up-regulating NuRD activity could lead to misregulation of genes and disease? In fact, to me it would seem that all the evidence points towards this being the case, and I think this would also be consistent with earlier work (much of it from the authors) where either over-expression or knock down of CHD4 results in altered chromosome structure. Could the dominant-negative and gain-of-function effects associated with the expression of CHD4 mutants in *Drosophila* (that the authors argue for in both the Introduction and Discussion) not involve either down- or up-regulation of NuRD? I would like to see some discussion of how CHD4 mutations could alter NuRD (not just CHD4) activity.

In the results, the authors use a gel shift assay to analyse binding of wild-type and mutant dMi-2 proteins to a mononucleosome using an electrophoretic mobility shift assay (Figure 2B). It was not clear to me that they are seeing true complex formation here. It looks as if that, with increasing protein concentration, everything aggregates and the proteins and nucleosomes no longer run into the gel. I think these experiments need to be repeated perhaps with a different type of gel (maybe agarose?) which allows them to demonstrate that the complexes run into the gel and show where the different complexes formed by the wild-type and mutant proteins run. Alternatively, if they cannot get good data using EMSA's perhaps it would be sufficient to rely on the other assays?

In summary, however, I think this is a nice paper and, with some revision, suitable for publication.

Reply to reviewers

We thank both reviewers for carefully considering our manuscript, for finding our data convincing and important and for their constructive criticism. We have prepared a revised version of our manuscript that contains additional experimental results and incorporates the reviewers' suggestions. Please, find a detailed point-by-point response to the reviewers' comments below:

Reviewers' comments:

Formatted: Font: Italic

Formatted: Font: Italic, English (U.K.)

Reviewer #1 (Remarks to the Author):

Formatted: Font: Italic

Formatted: Font: Italic, English (U.K.)

ATP-dependent chromatin remodelers, most notably the BAF complex, are frequently mutated in human cancers. The precise links between chromatin remodeling and cancer pathologies are not clear. Here, the authors mutated the Drosophila chromatin remodeler dMi-2 at residues that are frequently mutated in human CHD4 in endometrial carcinoma. The authors characterise the nucleosome remodelling properties of these dMi-2 mutants, including their nucleosome binding affinity, ATPase activity and nucleosome remodelling efficiency. Except the H1 198 mutation, all other mutations decrease nucleosome-remodeling activities in vitro. As in the case of ISWI, the data show that a decrease in remodeling activity does not always reflect a decrease in ATPase activity. The authors report that regions outside the ATPase domain, such as the chromodomains, are essential for the remodeling activity. Interestingly, the mutations point to a region, downstream to the ATPase domain, which can regulate the activity and direction of the remodeler without changing its ATPase activity. Finally, expression of these dMi-2 mutants in flies lead to ectopic branching or disconnected posterior crossveins in fly wings.

Formatted: Font: Italic

Formatted: Font: Italic, English (U.K.)

Formatted: Font: Italic

Formatted: Font: Italic, English (U.K.)

Authors have performed careful and comprehensive in vitro characterisation of the remodeling activities of the dMi-2 mutants. The data is convincing and important to understand the roles of CHD4 in human cancers. However, it is not clear if the changes in remodeling activities are the sole reason for the wing phenotype or the human cancer pathologies.

Formatted: Font: Italic

Formatted: Font: Italic, English (U.K.)

Formatted: English (U.K.)

We do not claim that the changes in remodeling activities of the mutants are the sole reason for the wing phenotype and human cancer pathologies. Indeed, we have noted that overexpression of wild type dMi-2 in the developing wing is already sufficient to elicit a gain-of-PCV phenotype (Figure 5). In the revised version we have included additional genetic experiments (see below and new Supplementary Figure 6) which support our findings. In addition, we explicitly state in the discussion that the cancer pathologies are the result of a combination of mutations and that it is unclear to what extent CHD4 mutations contribute:

"Cancer-associated dMi-2 mutants disrupt differentiation of wing structures

Formatted: English (U.K.)

Endometrial cancer arises when epithelial cells of the uterus become transformed and overproliferate. In serous endometrial carcinoma, the most aggressive form of this cancer, CHD4 mutations are most frequent and occur alongside mutations in tumour suppressor or oncogenes including TP53, PIC3CA, PPP2R1A or FBXW7⁶. The individual contribution of CHD4 mutations to malignant transformation is, therefore, difficult to assess."

Formatted: English (U.K.)

Formatted: English (U.K.)

The authors have already generated flies carrying these mutants, it should be relatively easy to provide more insights on the in vivo molecular mechanisms underlying the wing phenotypes. I therefore recommend publication with a revision to strengthen the in vivo data in the manuscript.

Formatted: Font: Italic

Formatted: Font: Italic, English (U.K.)

Major suggestions:

Formatted: Font: Italic

1. I suggest performing RNAseq or real time qPCR experiments in the wing disc or other tissues of the flies carrying these mutants, to test which dMi-2 bound genes are affected. Alternatively, authors can simply use real time qPCR to test TGF-beta/BMP signaling pathway components, as suggested in the manuscript. It will be very interesting if the affected genes turn out to reflect the abnormal gene expression observed in human cancers.

Formatted: Font: Italic, English (U.K.)

Formatted: Font: Italic

Formatted: Font: Italic, English (U.K.)

Formatted: English (U.K.)

This is a very good suggestion. We have tested expression of a set of eight genes encoding TGF-beta/BMP components in imaginal wing discs of transgenic flies by RT-qPCR. We have determined expression levels in wing imaginal discs overexpressing dMi-2 WT, a dMi-2 mutant with increased nucleosome remodeling activity *in vitro* (dMi-2 H1198Y) and a dMi-2 mutant with strongly reduced remodeling activity (dMi-2 R1164Q). Out of eight genes encoding TGFbeta/BMP signaling components tested, six displayed statistically significant increases in expression ranging from 1.3-fold to 2-fold relative to the control. These effects are mild but it is important to note that the *engrailed* driver used expresses the transgene exclusively in the posterior half of the wing imaginal disc, whereas RNA for RT-qPCR analysis was prepared from the entire disc. As a consequence, RNA levels measured reflect an average from transgene-expressing and non-expressing cells, thereby reducing the magnitude of gene expression changes. Intriguingly, upregulation of gene expression was detected irrespective of whether the dMi-2 transgene encoded a wild type enzyme, an enzyme with increased or an enzyme with decreased remodeling capacity. This correlates with the observation that expression of all dMi-2 trans genes resulted in gain-of-PCV phenotypes (Figure 5). It is conceivable that upregulation of TGFbeta/BMP signaling genes and the gain-of-PCV phenotype are the result of dMi-2 overexpression via a mechanism that is independent of dMi-2 remodeling activity.

Formatted: English (U.K.)

Expression of remodeling-defective dMi-2 generates additional loss-of-PCV phenotypes that were not observed upon expression of remodeling-competent dMi-2 (Figure 5). Given that overexpression of both remodeling-defective and remodeling-competent dMi-2 produces similar changes to the genes we have tested it is unlikely that these gene expression changes are causally related to the loss-of-PCV phenotype.

These new results are described on p.11 of the revised manuscript:

"PCV differentiation is very sensitive to changes in BMP/TGFbeta signaling. In order to assess if expression of genes encoding components these signaling pathways were derepressed by ectopic expression of dMi-2 we measured RNA levels in developing wing imaginal discs. We prepared RNA from wing discs from control larvae that carry a UAS-dMi-2 WT transgene but do not express GAL4 (*UAS-dMi-2 wt*) and larvae overexpressing dMi-2 WT (*en-GAL4>>UAS-dMi-2 wt*), the hyperactive remodeler dMi-2 H1196Y (*en-GAL4>>UAS-dMi-2 H1196Y*) or the remodeling-defective dMi-2 R1162Q mutant (*en-GAL4>>UAS-dMi-2 R1162Q*). We then performed RT-qPCR analysis (Supplementary Figure 7). Measuring dMi-2 mRNA levels confirmed overexpression of the three dMi-2 transgenes. We then tested expression of the BMP/TGFbeta genes *brk*, *Mad*, *Med*, *bi*, *cv*, *gbb*, *dpp* and *tkv*. Two of these genes (*brk* and *Med*) did not show a significant upregulation of RNA levels in wing discs expressing dMi-2 transgenes. The six remaining genes were upregulated between 1.25- and 2-fold with *tkv* showing the strongest effect. While these expression changes are mild it is worth noting that while RNA was prepared from whole wing imaginal discs the *engrailed* promoter drives expression of dMi-2 transgenes in the posterior half of the wing disc only, suggesting that the observed expression changes are an underestimation. Interestingly, genes encoding BMP/TGFbeta components were similarly upregulated by

Formatted: English (U.K.)

Formatted: English (U.K.)

Formatted: English (U.K.)

Formatted: English (U.K.)

Formatted: English (U.K.)

Formatted: English (U.K.)

Formatted: English (U.K.)

Formatted: English (U.K.)

Formatted: English (U.K.)

Formatted: English (U.K.)

Formatted: English (U.K.)

expression of WT, hyperactive and remodeling-defective dMi-2 proteins. This correlates with the observation that all three types of dMi-2 proteins generate gain of PCV phenotypes and indicates that this phenotype may be caused by expression of ectopic dMi-2 irrespective of its remodeling properties.

Formatted: English (U.K.)

Taken together, our results demonstrate that expression of cancer-derived dMi-2 point mutants in the background of endogenous, wild-type dMi-2 protein upregulates genes encoding BMP/TGFbeta components and is sufficient for derailing the differentiation of epithelial structures."

Formatted: English (U.K.)

Formatted: English (U.K.)

We agree with the reviewer that it would be very interesting if the changes in gene expression observed in the imaginal wing disc would reflect the abnormal gene expression observed in human cancers. Our results do not support such far reaching conclusions. However, it is important to stress that in endometrial cancer mutations in CHD4 (17% frequency) are generally accompanied by mutations in other genes such as TP53 (71%), PIK3CA (31%), PPP2R1A (25%) and others which also contribute to gene expression changes. In our experimental system we investigate the effects of mutations in dMi-2 only. While this informs us about the general effects of dMi-2 mutations on epithelial differentiation and gene expression our experimental system does not recapitulate endometrial cancer generation and progression.

It will also be advisable if the authors can map the in vivo nucleosome patterns on these affected genes in the mutants to understand the role of the remodeling activities.

Formatted: Font: Italic

Formatted: Font: Italic, English (U.K.)

Formatted: English (U.K.)

This would be a very good experiment in principle. However, we feel that this is outside of the scope of our manuscript. The experiment would require us to dissect wing discs and perform (e.g.) an MNase-ChIP experiment. Indeed, we have used such an assay in the past to study the effects of dMi-2 depletion on chromatin accessibility of ecdysone-responsive genes in S2 cells (Kreher *et al.*, *Nature Communications* 2017). However, S2 cells are a homogeneous cell population whereas wing discs are not. In fact, as explained above only the posterior half of the wing disc is expressing the transgene resulting in any effects on nucleosome positioning (and these effects are not dramatic even in S2 cells where dMi-2 has been almost completely depleted; Kreher *et al.* 2017) being diluted by signals from unaffected cells.

Provided that expressing the wild type Mi-2 alone already gives a wing phenotype due to overexpression, it may be better to express the wild-type/mutants in a heterozygous Mi-2 null-mutant background. It can imitate the heterozygous CHD4 mutant situations in human cancers and it hopefully also reduce the overall Mi-2 level to close to physiological endogenous level.

Formatted: Font: Italic

Formatted: Font: Italic, English (U.K.)

We agree with the reviewer that expression levels of the transgenes and the ratio between mutant and endogenous WT dMi-2 could influence the wing phenotypes observed. A heterozygous situation would indeed mimic the situation in cancer cells most closely. However, the generation of heterozygous flylines would be cumbersome and time consuming. We have chosen a simpler approach that addresses the same issue: The activity of GAL4 in flies is highly temperature dependent. We have conducted the fly experiments at the normal temperature of 29°C which affords optimal GAL4 activity (Duffy, *Genesis*, 2002). By contrast, at 18°C GAL4 activity is minimal (Duffy, *Genesis*, 2002). A wide range of transgene expression levels can be realised by simply varying the temperature at which the flies are kept. In contrast to transgene expression, expression of endogenous genes is largely independent of temperature. Accordingly, lowering the temperature does not only reduce transgene expression but also lowers the ratio between ectopic and endogenous dMi-2 protein, thereby bringing it closer to a heterozygous scenario.

Formatted: English (U.K.)

Formatted: English (U.K.)

Formatted: English (U.K.)

Formatted: English (U.K.)

Formatted: English (U.K.)

Formatted: English (U.K.)

We have repeated the UAS/GAL4 crosses at 18°C (new Supplementary Figure 6). As expected, this reduced the general penetrance of the phenotypes. By contrast, it did not alter the qualitative and quantitative differences between remodeling-competent and remodeling deficient dMi-2 mutants: First, expression of all WT and mutant dMi-2 transgenes generated gain-of-PCV phenotypes irrespective of whether the encoded enzyme was remodeling-competent or remodeling-defective. Second, dMi-2 H1196Y, which displays increased remodeling activity *in vitro*, still displayed the highest penetrance of gain-of-PCV phenotypes. In fact, the penetrance of dMi-2 H1196Y-induced gain-of-PCV phenotypes at 18°C (54%) was still greater than the penetrance of dMi-2 WT-induced phenotypes at 29°C (48%). Third, loss-of-PCV phenotypes, although detected in a much smaller proportion of flies, were still restricted to the expression remodeling-defective mutants. Thus, PCV phenotypes generated by expression of ectopic dMi-2 proteins are robust over a wide range of transgene expression levels and ratios of endogenous vs. ectopic dMi-2 expression levels.

Formatted: English (U.K.)

Formatted: English (U.K.)

Formatted: English (U.K.)

2. Authors reported an interesting observation that H1196Y mutation renders the remodeler with the same ATPase activity as the wild type Mi-2, but it becomes a better remodeler with no directionality to move the nucleosome to the center. It may suggest that the brace-1 region is responsible for measuring DNA length and give wild-type Mi-2 the directionality. Can the authors test binding between wt/H1196Y Mi-2 and mononucleosome with 10bp, 20bp, 30bp and 60bp overhang? Or test the ATPase activity with different overhang length? It may be that H1196Y can no longer distinguish a short vs a long DNA overhang. The authors can also provide other experiments that may explain the change in directionality of the enzyme.

Formatted: Font: Italic

Formatted: Font: Italic, English (U.K.)

Formatted: English (U.K.)

We thank the reviewer for this excellent suggestion. We have carried out both experiments (mononucleosome binding and ATPase assays). These results are included as the new Supplementary Figure 4 in the revised manuscript. We have tested binding to three types of nucleosomes: 0-77 nucleosomes (with a 77bp overhang on one side of the nucleosome), 0-44 nucleosomes and 0-22 nucleosomes. Both dMi-2 WT and dMi-2 H1196Y bind most efficiently to 0-77 nucleosomes. Binding becomes less efficient when the DNA overhang is shortened. Interestingly, binding of dMi-2 H1196Y is more strongly affected by the shortening of the DNA overhang than binding of dMi-2 WT: In agreement with our results with the 0-80 nucleosome shown in Figure 3C similar amounts of dMi-2 WT and dMi-2 H1196Y are required to shift the 0-77 nucleosome (lower panel). By contrast, compared to dMi-2 WT, twice the amount of dMi-2 H1196Y is required to shift the 0-22 nucleosome (upper panel). This supports the reviewer's hypothesis that dMi-2 H1196Y senses DNA length differently than dMi-2 WT and could contribute to the apparent loss of directionality in this mutant. We also measured the ATPase activities of dMi-2 WT and dMi-2 H1196Y in presence of the three types of mononucleosomes (new Supplementary Figure 4). This revealed no differences in ATPase activities. Finally, we also attempted to perform nucleosome sliding assays with the 0-22 nucleosome that displays the strongest difference in dMi-2 binding affinities. However, due to the short length of the overhang into which the histone octamer can be moved we failed to adequately resolve remodeled and non-remodeled nucleosomes with our gel system (*data not shown*).

Formatted: English (U.K.)

We have added this data as a new Supplementary Figure 4 and have described the results on p.9 of the revised manuscript:

"In addition, for some remodelers the direction of nucleosome sliding is determined by the length of the two DNA segments extending from the nucleosome core particle³⁴. In order to investigate if dMi-2

H1196Y senses the length of these DNA extensions differently than dMi-2 WT we tested binding to mononucleosomes with 22bp (0-22 nucleosome), 44bp (0-44 nucleosome) and 77bp (0-77bp nucleosome) extensions (Supplementary Figure 4A). Shortening the DNA segment extending from the core decreased binding affinity for both dMi-2 WT and for dMi-2 H1196Y. However, dMi-2 H1196Y was more strongly affected by this (Supplementary Figure 4A, compare upper and lower panels). These differences in binding affinity did not translate into differences in ATPase activity. Both dMi-2 WT and dMi-2 H1196Y ATPases were equally stimulated by all three types of mononucleosomes (Supplementary Figure 4B). Taken together, the results of our remodeling and nucleosome binding assays are consistent with the hypothesis that dMi-2 H1196Y senses DNA extending from the nucleosome core in a different manner. Thus, dMi-2 H1196Y might respond to the DNA sequence of nucleosomal DNA and/or the length of DNA extending from the nucleosome differently resulting in altered nucleosome positioning."

Formatted: English (U.K.)

Minor suggestions:

Formatted: Font: Italic

1. Simply increasing wild type Mi-2 protein amount is enough to cause a phenotype in the wing. This suggests that total protein level is important to interpret the result. The authors may provide western blot showing protein level of exogenous mutated Mi-2 and endogenous Mi-2 in these transgenic flies. The mutants are expressed from the same genomic loci, but the protein stability of different mutants may vary.

Formatted: Font: Italic, English (U.K.)

Formatted: Font: Italic

We agree with the reviewer that wing phenotypes might be influenced by expression levels. As explained above, we have carried out the UAS/GAL4 crosses at 18°C and 26°C which reflect low and high expression levels, respectively. While the penetrance of the phenotypes were indeed influenced by expression levels, the phenotype itself (gain-of-PCV versus loss-of-PCV) was not. Thus, we are confident that the phenotypes reflect *bona fide* properties of the dMi-2 enzymes tested.

Formatted: Font: Italic, English (U.K.)

Formatted: English (U.K.)

Formatted: English (U.K.)

Formatted: English (U.K.)

Formatted: English (U.K.)

Formatted: English (U.K.)

2. The title and abstract may cause misunderstanding. Almost the whole manuscript is based on the Drosophila protein Mi-2, but the abstract led me one think that the in vitro experiments were done with human proteins and human mutations. Drosophila was only mentioned in the genetic experiments. It would be good the make that clearer.

Formatted: Font: Italic

Formatted: Font: Italic, English (U.K.)

We did not intend to confuse the reader. We have now changed the title to read

Formatted: English (U.K.)

"Tumour-associated missense mutations in the dMi-2 ATPase alters nucleosome remodelling properties in a mutation-specific manner"

Formatted: English (U.K.)

We have also changed the abstract:

"A ATP-dependent chromatin remodellers are mutated in more than 20% of human cancers. The consequences of these mutations on their enzymatic activities and their *in vivo* function are poorly understood. **Here, we systematically characterise the effects of CHD4 mutations identified in endometrial carcinoma on the remodeling properties of dMi-2, the highly conserved Drosophila homolog of CHD4.** Mutations from different patients have surprisingly diverse defects on nucleosome binding, ATPase activity and nucleosome remodelling. Unexpectedly, we identify mutations that decrease as well as mutations that increase enzyme activity. Our results define both the chromodomains and a novel regulatory region adjacent to the ATPase domain as essential for nucleosome remodelling. Genetic experiments in *Drosophila* demonstrate that expression of cancer-derived dMi-2 mutants is sufficient to misregulate differentiation of epithelial wing structures and produces phenotypes that correlate with their nucleosome remodelling properties. Our results help to define the defects of CHD4 in cancer at the mechanistic level and provide the basis for the development of molecular approaches aimed at restoring their activity."

Formatted: English (U.K.)

Formatted: English (U.K.)

3. It is not clear to me if how L1215P disrupt nucleosome binding. Does the brace II region bind directly to nucleosomal DNA and L1215P disrupt the interaction with the nucleosome, or brace II bind to the ATPase core to inhibit its interaction with the nucleosome, and L1215P enhance the interaction with ATPase. Does the brace II region alone binds to nucleosome? Maybe the authors can elaborate.

Formatted: Font: Italic

Formatted: Font: Italic, English (U.K.)

Formatted: English (U.K.)

We currently do not have an answer to these questions. In related remodelers, such as Snf2, recent structural data suggests that the brace-I and brace-II regions are disordered and adopt their alpha-helical structures when the enzyme binds to the nucleosome (Liu *et al*, *Science*, 2017). In this case, the two helices protrude from the core1 domain and make direct contact with the core2 domain. From this structural work, there is no indication that brace-II directly contacts the nucleosomal DNA. The L1215P introduces a proline which is likely to disrupt the formation of the corresponding alpha-helix, potentially affecting the core1-core2 interaction in dMi-2/CHD4. Thus, we would hypothesise that this mutation indirectly reduces nucleosome binding by causing structural changes to the core1-core2 conformation. However, this hypothesis will have to be validated by analysing the structure of dMi-2/CHD4 itself, an undertaking that is outside of the scope of this manuscript.

Formatted: Font: Italic

4. It is unclear how the mutation in the PHD domain may affect the ATPase activity. In order to understand the function of this mutation, the authors may try to use nucleosomes that are methylated at H3K4 residue to perform their binding assay and ATPase assay. Only a minor fraction of endogenous histone octamer extracted from the cell contains H3K4me3. If it has been reported before in the literature, please elaborate.

Formatted: Font: Italic, English (U.K.)

Formatted: English (U.K.)

Indeed, analyses of the role of the CHD4 PHD fingers in nucleosome binding and ATPase function have been published by several labs (including the Laue, Mancini, Kutateladze and Mackay labs). The PHD fingers bind to the H3 tails of nucleosomes both individually and cooperatively in a modification sensitive manner where K4 methylation generally decreases and K9 methylation and K9 acetylation enhances binding. While we agree with the reviewer that it would be informative to test the activity of the PHD finger mutant on recombinant nucleosomes carrying defined methylation patterns we feel that this is not the focus of our study. However, we have changed the discussion of this mutant to better reflect the current state of knowledge by elaborating on the work of the above mentioned labs.

Formatted: English (U.K.)

The relevant section in the discussion now reads as follows.

"Mutation of the zinc binding C464 resulted in a mild reduction in dMi-2 ATPase and nucleosome remodelling activity. These results are consistent with the weak effect observed upon expression of this mutant in the developing wing (Figure 5) and with our previous finding that deletion of the PHD fingers of dMi-2 has only mild consequences for ATPase and nucleosome sliding activities²². A deletion of the PHD fingers to a CHD4 construct containing chromodomains and ATPase domain moderately increases ATPase and remodeling activities^{34,39}. In some cases these effects are influenced by the nucleosome methylation status³⁴. Indeed, the PHD fingers of CHD4 bind nucleosomes both individually and cooperatively in a H3 methylation-sensitive manner^{35,40}. The nucleosomes we have used in this study were assembled with histone octamers purified from *Drosophila* embryos. These contain a mixture of unmodified and methylated nucleosomes. It is, therefore, conceivable that the PHD finger mutation has a stronger effect in the context of the human CHD4 protein or in the presence of nucleosomes carrying defined H3 methylation patterns. We can also not exclude that this mutation affects intra- or intermolecular interactions that are not detected by our *in vitro* assays."

Formatted: English (U.K.)

Formatted: English (U.K.)

Formatted: English (U.K.)

Formatted: English (U.K.)

Formatted: English (U.K.)

Formatted: English (U.K.)

Formatted: English (U.K.)

In this section we cite the following papers:

34 Watson, A. A. *et al*. The PHD and chromo domains regulate the ATPase activity of the human chromatin remodeler CHD4. *J Mol Biol* **422**, 3-17, doi:10.1016/j.jmb.2012.04.031 (2012).

Formatted: English (U.K.)

35 Mansfield, R. E. *et al*. Plant homeodomain (PHD) fingers of CHD4 are histone H3-binding modules with preference for unmodified H3K4 and methylated H3K9. *J Biol Chem* **286**, 11779-11791, doi:10.1074/jbc.M110.208207 (2011).

Formatted: English (U.K.)

Formatted: English (U.K.)

Formatted: English (U.K.)

39 Morra, R., Lee, B. M., Shaw, H., Tuma, R. & Mancini, E. J. Concerted action of the PHD, chromo and motor domains regulates the human chromatin remodelling ATPase CHD4. *FEBS Lett* **586**, 2513-2521, doi:10.1016/j.febslet.2012.06.017 (2012).

Formatted: English (U.K.)

Formatted: English (U.K.)

40 Gatchalian, J. *et al.* Accessibility of the histone H3 tail in the nucleosome for binding of paired readers. *Nat Commun* **8**, 1489, doi:10.1038/s41467-017-01598-x (2017).

Formatted: English (U.K.)

Formatted: English (U.K.)

5. Figure 1 and 2 could be easily combined.

Formatted: Font: Italic

Formatted: Font: Italic, English (U.K.)

Formatted: English (U.K.)

This is in principle true. However, we prefer to devote one figure to each class of mutants and keep Figure 1 separate as an "overview". The total number of figures in our manuscript is not so large that we deem such a consolidation necessary.

Reviewer #2 (Remarks to the Author):

Formatted: Font: Italic

Formatted: Font: Italic, English (U.K.)

*Referee's Report for the manuscript entitled "Tumour-associated missense mutations in CHD4 ATPases alter their nucleosome remodelling properties in a patient-specific manner" by Kristina Kovač *et al.**

Formatted: Font: Italic

Formatted: Font: Italic, English (U.K.)

Formatted: Font: Italic

In this paper, the authors systematically characterise CHD4 mutations found in endometrial cancers. The interesting finding is that they identify mutations that increase, as well as decrease, the enzyme's activity. The in-vitro characterisation of the mutants is complemented by in-vivo experiments in Drosophila, which show that the expression of cancer-derived CHD4 mutants is sufficient to misregulate differentiation of epithelial structures. I thought this was a carefully designed and well thought out study, which is also clearly presented, and it deserves to be published.

Formatted: Font: Italic, English (U.K.)

Formatted: Font: Italic

Formatted: Font: Italic, English (U.K.)

Formatted: Font: Italic

Formatted: Font: Italic, English (U.K.)

Formatted: Font: Italic

Formatted: Font: Italic, English (U.K.)

Formatted: Font: Italic

Formatted: Font: Italic, English (U.K.)

In the Abstract, I think the authors should conclude with "Our results define the defects of CHD4 in cancer..."

Formatted: Font: Italic

Formatted: Font: Italic, English (U.K.)

Formatted: Font: Italic

We have followed the reviewer's suggestion and have changed the abstract accordingly.

Formatted: Font: Italic, English (U.K.)

Formatted: English (U.K.)

In the Introduction, the authors argue that CHD4 mutants do not contribute to disease by lowering CHD4 activity (haploinsufficiency). However, it is not clear to me why the results don't simply suggest that CHD4 levels are critical and that both down-regulating and up-regulating NuRD activity could lead to misregulation of genes and disease? In fact, to me it would seem that all the evidence points towards this being the case, and I think this would also be consistent with earlier work (much of it from the authors) where either over-expression or knock down of CHD4 results in altered chromosome structure.

Formatted: Font: Italic

Formatted: Font: Italic, English (U.K.)

Formatted: Font: Italic

Formatted: Font: Italic, English (U.K.)

The reviewer makes a very good point. Indeed, we cannot rule out that the CHD4 mutations contribute to disease simply by altering CHD4 activity levels. Our reasoning is based on the absence of any major deletions in the CHD4 gene in endometrial cancer (see LeGallo *et al.*, 2012) that would likewise lower CHD4 activity levels. Therefore, we favour the hypothesis that the point mutated dMi-2 enzymes exert dominant negative or gain of function effects. Nevertheless, we agree that both scenarios merit equal consideration. We have changed the relevant section in the introduction to:

Formatted: English (U.K.)

"However, it is not known whether and how disease-associated CHD4 mutations affect its enzymatic activities at the molecular level and how this impacts the epigenetic landscape, gene expression and genome stability. A analysis of the CHD4 mutation spectrum reveals two remarkable features: First, the majority of CHD4 alterations are missense mutations (89% in endometrial carcinoma)^{4,6,7}. Deletions, frameshift and nonsense mutations that would result in a complete loss or a truncated CHD4 protein are rare. Second, patients are heterozygous for CHD4 missense mutations and retain one wild type copy of CHD4^{4,6,7}. **It is possible that mutations in one of the two dMi-2 alleles lowers the total CHD4 activity in affected cells sufficiently to result in the misregulation of genes. In addition, CHD4 point mutants might contribute to disease not or not only by lowering overall CHD4 activity: Rather, the absence of**

Formatted: English (U.K.)

Formatted: English (U.K.)

Formatted: English (U.K.)

deletions, frameshift and nonsense mutations might suggest that defective CHD4 enzymes exert a dominant negative effect or that CHD4 mutations result in a gain of function."

Formatted: English (U.K.)

We have also changed a section in the discussion to give equal weight to both scenarios:

"It is clear that the majority of CHD4 mutations identified in endometrial cancers negatively impact ATPase and/or remodelling activity. **It is conceivable that a reduction of overall CHD4 activity in endometrial cancer cells contributes to cancerogenesis.** However, CHD4 gene deletions or frameshift mutations - which would likewise lower overall CHD4 activity - are very rare in endometrial cancer⁶. By contrast, missense mutations predominate (89%) arguing that **it might also be the mutated CHD4 protein itself that contributes to cancerogenesis via gain-of-function or dominant negative mechanisms.**"

Formatted: English (U.K.)

Formatted: English (U.K.)

Formatted: English (U.K.)

Could the dominant-negative and gain-of-function effects associated with the expression of CHD4 mutants in Drosophila (that the authors argue for in both the Introduction and Discussion) not involve either down- or up-regulation of NuRD? I would like to see some discussion of how CHD4 mutations could alter NuRD (not just CHD4) activity.

Formatted: Font: Italic

We thank the reviewer for his suggestion to include a discussion on the possible consequences of CHD4 mutations on NuRD activity. We have now included the following paragraph in the discussion:

Formatted: Font: Italic, English (U.K.)

Formatted: English (U.K.)

"The effects of CHD4 mutations might also manifest themselves by an altered activity of the NuRD complex. NuRD combines CHD4 nucleosome remodeling with HDAC1/2 histone deacetylase activities. Early work has suggested that remodeling is a pre-requisite for efficient nucleosome deacetylation². Thus, altered CHD4 activity is likely to have an impact not only on nucleosome positioning but also on histone acetylation."

Formatted: English (U.K.)

Formatted: English (U.K.)

In the results, the authors use a gel shift assay to analyse binding of wild-type and mutant dMi-2 proteins to a mononucleosome using an electrophoretic mobility shift assay (Figure 2B). It was not clear to me that they are seeing true complex formation here. It looks as if that, with increasing protein concentration, everything aggregates and the proteins and nucleosomes no longer run into the gel. I think these experiments need to be repeated perhaps with a different type of gel (maybe agarose?) which allows them to demonstrate that the complexes run into the gel and show where the different complexes formed by the wild-type and mutant proteins run. Alternatively, if they cannot get good data using EMSA's, perhaps it would be sufficient to rely on the other assays?

Formatted: Font: Italic

As explained in our response to reviewer #1 above, we have now performed additional bandshifts with a series of nucleosomes with shorter DNA overhangs (new Supplementary Figure 4). In particular, the 0-22 and 0-44 nucleosomes are small enough to allow the dMi-2/nucleosome complex to enter the gel (upper and middle panel). We conclude that dMi-2 does form a proper complex with the nucleosome and does not aggregate under our bandshift conditions. However, when the DNA overhang gets too long (as in the 0-80 nucleosomes used in Figures 2, 3 and 4) this complex is too large to enter the gel.

Formatted: Font: Italic, English (U.K.)

Formatted: Font: Italic

Formatted: Font: Italic, English (U.K.)

Formatted: English (U.K.)

In summary, however, I think this is a nice paper and, with some revision, suitable for publication.

Formatted: Font: Italic

Formatted: Font: Italic, English (U.K.)

Reviewers' Comments:

Reviewer #1:

Remarks to the Author:

Overall, authors have address the major points I had raised in my original review. Therefore, I support publication of this revised manuscript.

More specifically:

The authors have provided qPCR data to support their claim and sufficiently responded to my comment.

It is a pity that the authors cannot provide the in vivo nucleosome pattern in their mutants, but the authors have explained the technical challenge, and it is understandable.

The author have addressed the concern on the ratio of WT/mutant expression level, by modifying the temperature and hence the expression level. The concern is sufficiently addressed.

The author have performed the in vitro assay with different size of overhang as suggested. The data sheds new light on why H1196Y may behave differently than the WT. The outcome is informative and interesting.

The authors have sufficiently addressed minor concerns 1, 2, 4, as well as explained minor concern 3, 5 adequately.

Reviewer #2:

Remarks to the Author:

I have read the Response to Reviewers and the new manuscript and am very happy to recommend publication.